# Role of Receptors in Relation to Plaques and Tangles in Alzheimer’s Disease Pathology

**DOI:** 10.3390/ijms222312987

**Published:** 2021-11-30

**Authors:** Kavita Sharma, Samjhana Pradhan, Lawrence K. Duffy, Sabina Yeasmin, Nirajan Bhattarai, Marvin K. Schulte

**Affiliations:** 1Biomedical and Pharmaceutical Sciences, College of Pharmacy, Kasiska Division of Health Sciences, Pocatello, ID 83209, USA; sharkum2@isu.edu (K.S.); sabinayeasmin@isu.edu (S.Y.); nirajanbhattarai@isu.edu (N.B.); 2School of Chemical Engineering, Yeungnam University, Gyeongsan-si 38541, Korea; samjhanapradhan313@gmail.com; 3Department of Chemistry and Biochemistry, University of Alaska Fairbanks, Fairbanks, AK 99709, USA; lkduffy@alaska.edu

**Keywords:** Alzheimer’s disease, amyloid beta, G-protein-coupled receptors, nicotinic acetylcholine receptors, N-methyl D-aspartate receptors, γ-aminobutyric acid (GABA)

## Abstract

Despite the identification of Aβ plaques and NFTs as biomarkers for Alzheimer’s disease (AD) pathology, therapeutic interventions remain elusive, with neither an absolute prophylactic nor a curative medication available to impede the progression of AD presently available. Current approaches focus on symptomatic treatments to maintain AD patients’ mental stability and behavioral symptoms by decreasing neuronal degeneration; however, the complexity of AD pathology requires a wide range of therapeutic approaches for both preventive and curative treatments. In this regard, this review summarizes the role of receptors as a potential target for treating AD and focuses on the path of major receptors which are responsible for AD progression. This review gives an overall idea centering on major receptors, their agonist and antagonist and future prospects of viral mimicry in AD pathology. This article aims to provide researchers and developers a comprehensive idea about the different receptors involved in AD pathogenesis that may lead to finding a new therapeutic strategy to treat AD.

## 1. Introduction

An estimated 50 million people live with dementia worldwide. Current projections indicate a three-fold increase by 2050, affecting an estimated 152 million people [1]. Living with dementia is both very difficult and expensive. The 2018 World Alzheimer Report approximated annual global costs at a trillion U.S. dollars, representing 1.16 % global GDP; however, this value is not limited and is expected to double by 2030 [2]. 

Alzheimer’s disease (AD), the most common neurodegenerative disease, is prevalent among people over the age of 65. AD patients exhibit neurocognitive disorders, incipient cognitive decline, and memory loss, along with psychological and physical trauma [3]. Causative factors and disease etiology remain unclear [4]. After an international research effort, several hypotheses describing AD pathology have been discussed, but only a few have been well reported, including the β-amyloid (Aβ) cascade hypothesis [5], the cholinergic neuron damage hypothesis [6], and the Tau hypothesis [7]. Nonetheless, these theories have been unable to identify a unifying treatment for AD pathology and the disease remains complex phenomenon. 

Multiple approaches to define the mechanism responsible for AD pathology have been proposed, including Aβ cascade, tau and cholinergic hypotheses, increased oxidative stress [8] and metabolic diseases, including type 2 diabetes with high-fat diet consumption [9] have also been associated with AD disease. Some researchers believe amyloid-β is an oxidative stress response (ROS) in which a hydroxyl radical binds to the amyloid-β that may later lead to oxidative damage of the Aβ peptide and its surrounding molecules (e.g., proteins, lipids, DNA) [10].

A number of studies have targeted and repurposed the observed AD pathological sites (e.g., receptors and gene mutations) as potential therapeutic tools for AD treatment. Receptors detect chemical signals and/or physical stimuli. They bind with signal molecules and transmit instructions through a series of molecular events resulting in a cellular response. In AD patients, the cell signaling mechanism is considered to be dysfunctional. In this regard, receptors have a prominent relation with AD pathology, thereby, serving as a potential target for treating the disease [11,12]. Likewise, researchers have also identified several genes as a therapeutic target for AD pathogenesis. In particular to the genetic mutation observed in APP, PSEN1, or PSEN2 genes that are associated with early onset of Alzheimer’s disease (EOAD) [13]. Regardless, receptors are our prime focus in this review, and we have attempted to highlight their specific functions in the realm of AD pathology.

Amyloid-β (Aβ) plaque and neurofibrillary tangles (NFTs) in AD patients have been a hallmark indicator for AD’s neuropathological diagnosis. The amyloid cascade hypothesis, earlier studies suggested that AD neurodegeneration AD is caused by the formation of extracellular Aβ peptide deposited in senile plaques, followed by the intracellular accumulation of malformed tau protein in NFTs [14]. The aggregated form of the insoluble amyloid-β (Aβ) peptide is considered the driving force of AD pathogenesis that results in the formation of NFTs containing tau protein. This occurs due to the imbalance between Aβ production and Aβ clearance [15]. The deposition of the protein fragments of Aβ disrupt the communication between the cells of the brain, thereby triggering the immune cells. This action triggers inflammation of the brain cell causing death in the final stage. The identification of amyloid beta peptide in AD brains have led researchers to design experimental drugs targeting these protein components. The main objective is either to reduce the production of Aβ peptide or enhance their removal from the brain and prevent their aggregation [16]. Nonetheless, these approaches have futile in developing drugs that eliminate Aβ peptide as a treatment to mitigate the behavioral effects found in the disease’s late stages. Later AD pathogenesis studies have suggested that the senile plaques may not be directly involved neuron and synapse loss. Rather, the soluble oligomers of Aβ as diffusible entities, neither monomers nor insoluble amyloid fibrils, are considered responsible for synaptic failure and memory impairment in the brains of both AD animal models and patients [17,18].

Aβ peptide aggregation has also been linked with dysfunction in the cholinergic system. The cholinergic system involves nicotinic acetylcholine receptors (nAChRs) and muscarinic acetylcholine receptors (mAChRs) that bind to the neurotransmitter, acetylcholine (ACh), for signal transduction. The cholinergic hypothesis suggests that the cholinergic system plays a critical role in cognitive function. Cholinergic system dysfunction related to ACh synthesis and the death of cholinergic neurons stimulates age-related cognitive impairment resulting in extensive presynaptic cholinergic denervation. Administrating cholinesterase inhibitors to increase the availability of ACh and delay synaptic degradation in the brain is being used as a therapeutic approach to treat AD [19]. 

AD’s intricate pathological mechanisms have led to multifaceted hypotheses and experimental approaches. While the cholinergic hypothesis may be within the purview of the AD pathology, ACh is not the sole neurotransmitter responsible for denervation. There are other possible neurotransmitters and pathways that contribute to AD pathology. The mechanism remains unclear. Amyloid’s β-sheet structure binds to multiple sites and it is assumed that the Aβ peptide plays a role in decreasing the release of presynaptic ACh and impeding the coupling of postsynaptic muscarinic acetylcholine receptors (mAChR) to G proteins. This action reduces both signal transduction and APP level modulation and, as a result, simultaneously increases the production of Aβ peptide and further decreases ACh levels [11]. 

NFTs are insoluble proteins formed due to hyper phosphorylation [20]. Tau, a phosphoprotein found in central nervous system (CNS) neurons, is responsible for maintaining the stability of microtubules (MTs) found in the neuron’s axon and are responsible for transporting nutrients and information throughout the neurons [21,22]. Tau protein structures can be phosphorylated at various sites by protein kinases. In AD patients, hyper phosphorylated forms of tau protein destabilize the MTs and form aggregates or tangles of paired helical filaments (PHFs). The tangles build into intracellular masses of insoluble NFTs that disrupt cytoplasm function and axonal transport, resulting in neuronal cell death [23]. The role of tau pathology in neuronal cell death is considered a significant part of the dementia and various studies have focused therapeutic development to directly inhibit tau phosphorylation, aggregation and propagation, and microtubule stabilization [24].

The amyloid hypothesis has dominated AD research for decades. Clinical trials have been relatively unsuccessful and alternate perspectives on AD pathology etiology are now emerging. One perspective suggests that viral infection is seeded in the brain and implicates Herpes Simplex Virus Type 1 in early Alzheimer’s disease progression [25]. The prospect of neurological disorder prompted by virus exposure is far-reaching, and addresses unexplored linkages between viral infection and neurological disease. In this context, our group has been working on a novel approach that examines viral mimicry in AD pathology. The foundation of this hypothesis lies in the structural and sequence similarities between Ly6 family proteins and virus glycoproteins domains. The potential mechanism for Alzheimer’s disease based on structural similarity of viral glycoproteins binding epitopes to the LY6 family of proteins is certainly a novel approach associating viral infection with neurodegeneration. 

Our previous work provides evidence for virus receptor interaction and host manipulation using rabies virus glycoprotein homologous. We found that snake toxins inhibit nicotinic acetylcholine receptor (nAChRs) in the central nervous system [26]. Viruses have been shown to contain coat proteins that include the three-finger toxin (3FTx) domain, the prototypical rabies virus glycoprotein (RGP), and the gp-120 protein in the HIV virus [27]. GP-120 interaction with nAChRs may play a role in the development of HIV related dementia, which shows similarities to Alzheimer’s disease, including the formation of amyloid plaques [28].

Our hypothesis suggests specific molecular mechanisms, based on molecular mimicry of viral proteins to endogenous LY6 proteins, lead to neurodegeneration. While untested, this hypothesis has significant potential for producing a new line of inquiry in Alzheimer’s disease progression, treatment, and the potential development of early diagnostic tests. This review endeavors to provide a comprehensive description of the receptors responsible for regulating memory and cognitive performance in a healthy brain. Likewise, we address the ramifications amyloid plaques and tau proteins have on the working capacity of these receptors, the impact on synaptic signaling and cognitive functions that lead to neurodegeneration and cognitive impairments. Understanding the receptors’ mechanism is essential to unraveling AD pathophysiology and designing interventions that delineate prospective pharmacological targets to treat AD engendered neuronal dysfunction. 

## 2. Amyloid Beta (Aβ) Formation

One of the prime suspects in AD pathology, β-amyloid is a major component of amyloid senile plaques derived from the proteolytic action of proteases such as β-secretase and γ-secretase on amyloid-β precursor protein (APP) [29,30]. APP is an integral membrane protein located at the synapses of brain cells. There, it functions as a cell surface receptor and regulates synapse formation, neurite growth, neuronal adhesion and axonogenesis [31]. Many researchers believe the APP molecule forms Aβ peptide, a 37 to 49 amino acid residue that comprises the core section of the amyloid plaque [30,31]. 

In human neurons, cleavage of APP is a complex procedure and occurs via several pathways. Previous studies have reported that APP cleavage takes place by two different routes, the amyloidogenic and non-amyloidogenic pathways as shown in Figure 1. The amyloidogenic process involves the enzymatic activity of β- and γ-secretases on APP to produce the extracellular, insoluble amyloid plaques. The action of α-secretases on APP, generates the soluble amyloid precursor protein α (sAPPα) fragment with N-terminal and a C-terminal fragment, C83, (CTF83) accounting for the non-amyloidogenic processing. CTF83 is further cleaved by γ-secretase to produce p3, a truncated Aβ fragment and APP intracellular domain (AICD) [32]. The processing of α-secretase for the release of sAPPα, in the non-amyloidogenic pathway, is regulated by phosphatidylinositol 3 kinase (PI3K) [33], mitogen activated protein kinase (MAPK) extracellular signal regulated kinase (ERK) [34], protein kinase C (PKC) [35], and cyclic AMP–protein kinase A (cAMP-PKA) [36]. 

sAPP-α has been reported to have significant physiological function in neurogenesis, neuroprotection, memory formation, synaptic plasticity, and neurite growth [37]. Ishida et. al. demonstrated that sAPP-α shifts the frequency dependence for induction of long-term depression (LTD) and increases long-term potentiation (LTP) synaptic transmission in rat hippocampal slices [38]. In AD patient’s cerebro spinal fluid (CSF), Sennvik et al. found a decreased level of α-secretase-cleaved sAPP and total sAPP activity, suggesting that the decreased secretase activity may contribute to the development of AD [39]. sAPP-α has also been reported to regulate β–secretase activity and amyloid-β generation. In transgenic mice, the overexpression of sAPP-α decreases the level of Aβ plaques. These soluble APP-α directly interact with β-site APP-cleaving enzyme, BACE and ameliorate the APP imbalance processing that can lead to AD pathogenesis [40].

In a similar manner, APP is fragmented first by β-site APP cleaving enzyme (BACE) in the amyloidogenic or β-secretase pathway. This action generates N-terminal soluble fragments of amyloid precursor protein β (sAPPβ) and the C-terminal APP fragment C99 (CTF99) [41]. CTF99 is then cleaved by γ-secretase, composed of presenilin, Aph-1, Nicastrin and Pen-2 proteins [42], to release AICD along with Aβ peptide of length ranging from 37 to 43 amino acids units. Within this range, the Aβ peptides, Aβ40 and Aβ42, are considered the predominant constituent of extracellular insoluble amyloid beta plaques and have neurotoxic properties [43]. Comparatively, Aβ42 with longer lengths is highly susceptible to aggregate forming amyloid plaques and is better able to mediate neurotoxicity than Aβ40. This action suggests that presenilin mutation is the causative factor boosting Aβ42 production ratio compared to Aβ40 [43,44]. The fragmented Aβ40/42 peptides have been shown to be responsible for several downstream pathways related to AD. However, there is no correlation between levels of cortical plaques and AD-related cognitive impairment. The relationship between Aβ and neurotoxicity in AD pathology is poorly understood. 

Recent FDA approval of the drug aducanumab has been both unprecedented and controversial. Targeting the removal of Aβ plaque, the drug is one of the first therapies to address the causative factor rather than treating the symptoms of AD pathology. Approved for its ability to reduce the levels of senile plaques, the manufacturer, Biogen (Cambridge, Massachusetts, U.S. city, claims that infusion of the drug’s highest dose has slowed cognition decline in a group of AD patients, yet Boogen’s claims regarding aducanumab remain inconclusive [45].

Due to the lack of evidence that equates senile plaques with neuronal loss and cognitive decline in AD pathology, several studies have alternatively proposed soluble intracellular Aβ oligomers (AβOs) as the primary factor of AD pathogenesis. The soluble oligomers of Aβ or amyloid-β-derived diffused ligands (ADDLs) are believed to induce aberration in synapse composition, shape, and abundance [46]. These oligomers are considered to have a role in early synaptic pathology in AD [47]. Supporting the ADDL hypothesis, Catalano et al. has suggested that overproduction of Aβ42 induces oligomerization of Aβ42 and ADDL formation. ADDLs bind to neuronal receptors that lead to memory impairment and chronic synaptic dysfunction, ultimately resulting in the characteristic dementia of Alzheimer’s disease [48]. 

Similarly, the extracellular AβOs interact with glutamate receptors at postsynaptic membrane, resulting in dysregulation of calcium influx to impair LTP and enhance LTD [18,49]. It has also been reported that intracellular AβO originates from the internalization of extracellular Aβ through Aβ internalization receptors. Receptors such as low-density lipoprotein receptor-related proteins, apolipoprotein E receptors, and α-7 nicotinic acetylcholine receptor participate in the Aβ internalization process. Potential targets for AD therapies may be identified as our understanding of internalization receptors improve [50]. Mechanistically, the interaction of AβO with the abovementioned receptors is not fully understood and requires further elucidation. Using receptors to develop AD therapies is an emerging effort driving our understanding of [50,51].

## 3. G-Protein-Coupled Receptors

G-protein-coupled receptors (GPCRs) are membrane proteins that form the largest and most diverse group of cell surface receptors in eukaryotes. These receptors bind with G-proteins in the plasma membrane and activate cellular responses by detecting molecules from outside of the cell. GPCR is made up of a single polypeptide arranged in a globular shape with extracellular N-terminus and intracellular C-terminus encircling the plasma membrane seven times and hence are also known as seven-transmembrane (7TM) receptors [51]. 

G-proteins are heterotrimeric with three subunits made up of α, β and γ subunits. There are four main classes of G proteins: Gi that inhibits adenylyl cyclase; Gs that activates adenylyl cyclase; Gq that activates phospholipase C; and G12 and G13, of unknown function. Gα-subunit binds either to guanosine triphosphate or guanosine diphosphate, depending on the nature of its activity. In the absence of signaling molecules (inactive stage), GDP attaches to the Gα-subunit, and the whole Gα-GDP complex is bound to a nearby GPCR. However, on binding with any extracellular signals (active stage), a conformational change is observed in GCPR that leads to G-protein activation. This process replaces GDP with GTP in the Gα subunit, and eventually dissociates into Gα-GTP complex and Gβγ dimer, both attached to the plasma membrane. In its active form, both the complex and dimer structures relay messages within the cell by interacting with other membrane proteins (second messengers) such as cyclic adenosine monophosphate (cAMP), diacylglycerol (DAG), and inositol-1,4,5-triphosphate (IP3) [52].

The activated G-protein communicates a large number of second messenger small molecules, especially adenylyl and phospholipase C, to initiate and coordinate intracellular signaling pathways (Figure 2). When directed by the activated Gα-GTP complex, adenylyl cyclase acts as a catalyst for the synthesis of cAMP from ATP molecules, and phospholipase C acts as a catalyst for the synthesis of DAG and IP3 from the membrane lipid phosphatidyl inositol. In all mammals, cAMP has an important role in biological processes such as regulation of neurotransmitter synthesis, gene regulation, growth factors, immune function, hormone response, sensory input, and nerve transmission. Similarly, the activated Gα proteins are also involved in the activation of the mitogen-activated protein kinase (MAPK) and phosphatidylinositol 3-kinase (PI3K) pathways. Activation of PI3K and MAPK pathways result in the phosphorylation of protein kinase B (PKB) and extracellular signal-regulated kinases (ERKs), respectively. Eventually, activated PKB will phosphorylate and inhibit the action of GSK3β (glycogen synthase kinase 3β), a major kinase in the brain [53].

Many physiological processes result from a signal (e.g., lipids, peptides, proteins, neurotransmitters, or stimuli from light and odors) binding to the GPCR. Through this process, receptors are activated and initiate the production of a number of secondary messengers that regulate bodily functions such as sensation, growth, pain, hormone response, immune response, mood and synaptic transmission making GPCRs a priority target for clinical drug development [54]. Nonetheless, due to its complex signaling pathways, studies have not yet identified these receptors as an AD drug development target.

GPCRs are involved throughout AD’s progression and the GPCR superfamily includes a wide variety of monoamine neurotransmitters, (e.g., serotonin (5-HT), dopamine, γ-aminobutyric acid (GABA), and norepinephrine (NE)), disrupted by AD pathology. Thathiah et al. describes the GPCR receptors as direct modulators of α-, β- and γ-secretases, APP (amyloid precursor protein) processing, and amyloid-β degradation, and have suggests that GPCRs could serve as a prospective therapeutic target for treating AD [11,55]. GPCRs regulate the sequential cleavage of APP by α-, β- and γ-secretases ultimately determining the degree of Aβ peptide generation and its direct and indirect influence on GPCR function [56]. This is accomplished by either stimulating GPCR or producing GPCR-based drug through the modulation of signal paths. For instance, PKC, cAMP–PKA, MAPK–ERK, and PI3K, could provide an alternative measure for neuroprotection, memory, and cognitive enhancement in AD patients.

Considered a good target for the development of novel therapeutics to treat AD pathology, GPCRs are involved in several pivotal neurotransmission and signaling processes such as glutamatergic, cholinergic, serotonergic and adrenergic that are interrupted in AD. These receptors regulate the formation of amyloid beta peptides [11,55], influence calcium homeostasis, involve memory functions [57], modulate microglial activation, and leading to the generation of Aβ peptide and APP cleavage in AD brains [58]. Subsequently, signal transduction routes interference, involves PKC, cAMP–PKA, MAPK–ERK, and PI3K, in numerous neurotransmitter systems, and could provide an alternative measure for AD patient neuroprotection, memory, and cognitive enhancement.

In vivo and in vitro studies performed on an AD transgenic mouse model and murine microglial cells have demonstrated that the effect of Aβ peptide on GPCR is dose dependent [59]. At a concentration of 5 µM and above, Aβ directly triggers the release of tumor necrosis factor α (TNF-α), whereas, at a sub threshold dose of Aβ, release is indirect and modulated by certain GPCR activators. The treatment sub threshold level of Aβ reduces GPCR kinase-2/5 (GRK2/5) in the plasma membrane leading to retarded GPCR desensitization, prolonged GPCR signaling, and cellular hyperactivity to GPCR agonists. The in vivo GRK dysfunction occurs at prodromal and early stages of AD [59]. GPCR signaling also monitors pre- and postsynaptic neurotransmission, which can lead to changes in synaptic plasticity, including long-term depression (LTD), long-term potentiation (LTP), depotentiation (reversal of LTP), and presynaptic vesicle release potential [60]. In this context, modulation of GPCR signaling and transduction is regulated by a family of RGS (a regulator of G-protein signaling) proteins. These protein components act as a negative regulator of GPCR-G protein signaling by acting as GAPs (GTPase activating proteins) for Gα subunits. RGS proteins play a critical role in neuronal signaling and are considered potential therapeutic targets for various neurological disorders including AD pathogenesis [60].

## 4. N-Methyl D-Aspartate (NMDA) Receptors

N-methyl-d-aspartate receptor (NMDAR) is a ligand-gated ionotropic glutamate receptor that selectively binds with NMDA for neurotransmission. This glutamatergic receptor consists of two glycine-binding subunits (i.e., GluN1), two glutamate-binding subunits (i.e., GluN2A, GluN2B, GluN2C, and GluN2D), a combination of a GluN2 subunit and glycine-binding GluN3 subunit (i.e., GluN3A or GluN3B), or two GluN3 subunits. Thus, in NMDR, the binding of two different neurotransmitters, glutamate and glycine, is essential for the activation of glutamate-gated ion channel [61]. In addition, the activation for NMDAR is also largely dependent on the removal of the extracellular Mg^2+^ block (voltage dependent) through a strong membrane depolarization, that subsequently allows an influx of Ca^2+^ to initiate synaptic signaling that is essential for learning and memory functions [62]. NMDAR are expressed in the CNS where they are responsible for transmitting the Ca^2+^ permeable excitatory neuronal signals and are considered to have a significant role in synaptic plasticity and memory functions. NMDAR dysfunction, including receptor inactivation and Ca^2+^flux of, is associated with learning and memory impairments, which are characteristic features of age-related neurodegenerative disease such as AD [12]. Understanding NMDAR’s role at the central synapses may prove beneficial for determining therapeutic targets for reversing the memory disorders and synaptic functioning associated with AD pathology. 

In the mammalian CNS, the excitatory glutamate neurotransmitter is predominant and acts on a number of glutamate neurotransmitters, including NMDARs. While synaptic NMDARs are crucial for normal brain functioning, extra-synaptic NMDARs and over activation of these excitatory receptors leading to excess Ca^2+^ influx in the cells, is responsible for neuronal excitotoxicity [63]. Similarly, dysfunctional glutamatergic signaling, involving Ca^2+^ homeostasis via NMDAR, has a direct link to AD pathology. Aβ plaques and hyper phosphorylated tau proteins are considered responsible for disrupting glutamatergic tripartite synapse functioning. The tripartite synapse consists of presynaptic and postsynaptic terminals along with the glial cells. The synapse receptor proteins modulate the extracellular glutamate levels and are considered deregulated in AD pathology [64]. 

Amyloid-β peptide (Aβ), one component of the extracellular amyloid plaque, is a hallmark of Alzheimer’s disease (AD). Proponents of the amyloid hypothesis argue that the accumulation of Aβ in the brain is AD’s driving factor [65]. An increase of Aβ accumulation leads to NMDA-induced synaptic dysfunction via the activation of NMDAR’s extra synaptic NR2B subunit, fostering NMDAR endocytosis at the synapse while also shrinking glutamate reuptake and promoting glutamate spillover [66]. The aftermath of NMDAR endocytosis is attenuation of NMDAR mediated Ca^2+^ influx at the synapse and spine head. In addition, Aβ is also shown to hinder the supply of AMPA receptors to post synaptic neurons [66]. Paradoxically, Aβ excitotoxicity is actuated by over activation of extracellular NR2B due to inhibition of intracellular glutamate reuptake. Further, it has been shown that Aβ promotes release of glutamate from astrocytes. All these phenomena promote the sustained influx of Ca^2+^ through NMDAR rendering an excitotoxic effect [67]. Interestingly, amyloid precursor protein (APP) processing is expedited upon NMDA receptor activation at the synapse, whereas Aβ is synthesized by beta-secretase cleavage of APP on extra-synaptic activation.

NMDA receptor activation occurs when endogenous agonist glutamate NR2 family subunits bind at the NR1 subunit. Simultaneously, it co activates glycine, and releases the magnesium block by membrane depolarization. D-Serine, the other endogenous ligand, found abundantly in astrocytes, shares the same glycine binding site with somewhat similar potency. Interestingly, D-serine seems to be the dominant coactivator of NMDA induced neurotoxicity compared to glycine. The magnesium and S-nitrosylation sites are two prominent modulatory sites located inside the ion channel and extracellular amino-acid terminal domain (ATD), respectively. ATD offers allosteric regulation of ion channel open probability, receptor deactivation speed. Selective calcium permeability and the magnesium blockade are mediated by arginine residue present in the pore lining of the transmembrane loop (M2). S1 and S2, the two discontinuous ligand binding domain, constitute ligand binding location. Upon ligand binding and subsequent receptor activation, the Mg^2+^ block is released, the channel opens and the influx of Ca^2+^, Na^+^ and K+ occurs inside the neuron [68,69]. 

Harnessing the neuroprotective effect of the NMDAR blockade has long been pursued as a target to address various neurological diseases, including Alzheimer’s disease; however, harsh side effects have resulted in failed clinical trials. Envisioning future NMDAR drug discovery targets, Lipton put forward a conceptual framework for tolerated antagonists, explicitly negating the possibility of competitive antagonists and arguing that the competitive blockade may also affect healthy brain function involving glutamate or glycine activation of NMDA [67]. In fact, high extracellular glutamate or glycine accumulation over time displaces antagonists, thus preferentially altering healthy brain rather than pathological physiology. An uncompetitive open channel blockade emphasizing on-off rate was presented as an alternative. Succinctly, an uncompetitive antagonist offers greater inhibition to agonists at higher concentrations than lower concentrations. In addition, an optimal off rate that is slower than Mg^2+^ yet faster than MK-801′s off rate enables the ligand to achieve an excessively open state, compared to the receptor, under normal transmission. One exception, Memantine, has a proven clinical safety profile [68], is well tolerated, is an effective treatment of moderate to severe AD [70], and fits well in this paradigm. Memantine’s success may be due to the drug’s selectivity to extra synaptic NMDAR [67], its low-affinity/rapid off rate, and uncompetitive channel blockade [70]. 

An uncompetitive NMDAR antagonist, memantine is thought to slow neuronal cell death observed in AD by blocking the action of the NMDA-receptor ultimately inhibiting the over activation of glutamatergic neurotransmission, particularly in relation to calcium influx resulting from excessive NMDAR stimulation [71,72]. Memantine is an amantadine derived compound and the amino group in the adamantane ring binds at or in the vicinity of the Mg^2+^ binding site in the NMDA-gated channel [73]. Besides this major binding site, another memantine binding site has also been reported. This superficial site has low affinity at the channel vestibule and also serves as a memantine blocking site [74]. 

Memantine is used along with acetylcholine esterase inhibitors for treating moderate to severe AD cases. It has the ability to block the action of excessive NMDAR; however, due to its low affinity binding power it can be easily replaced, subsequently releasing the NMDAR blocking action [75]. This scenario demonstrates the importance of identifying molecular devices that mimic the action of NMDAR blocking tools and their ability to modulate and fine tune the NMDA-receptor function. As NMDAR serves as the key factor in controlling synaptic plasticity and memory performances, such process anticipates as an effective therapeutic approach for the improvement of cognitive and learning functions in AD pathology and other brain disorders. 

## 5. α-Amino-3-hydroxy-5-methyl-4-isoxazolepropionic Acid Receptor (AMPAR)

AMPA receptors are the ionotropic glutamate receptors formed by the combination of four subunits, GluA1-4 that are encoded by GRIA1-4 genes. These glutamate receptors mediate a rapid post synaptic transmission in CNS with a major role in long term synaptic plasticity such as LTP and LTD and are also involved in long term memory retrieval [76]. The biophysical property of AMPARs, involving the subunit composition, is instrumental in understanding the mode of AMPAR trafficking and synaptic plasticity. The most recurring AMPAR is the heteromeric composition of GluA1A2, which is dominant at CA1 cell synapses and is the primary mediator of synaptic plasticity [77]. Similarly, the dysfunction in the expression of all the four subunits of AMPAR, have also been reported as a cause of neurodevelopmental disorders (NDDs) [78]. Thus, in this instance, it is important to highlight the involvement of AMPARs in the development of AD pathology, given its critical role in synaptic transmission and NDDs.

AMPARs are considered as a susceptible target of AD pathophysiology. The high levels of Aβ impairs the functional AMPAR and disrupts the excitatory synaptic transmission [79], where it has been reported that the cognitive impairment involves an age related downscaling of post synaptic AMPAR function in double knock in mice (2 X KI) with human mutation of gene for APP and presenilin [80]. The reduction of AMPAR indicates that there are changes in the early phase of pathological molecular level in AD. In this regard, Zhang et al. suggested that the reduction in the level of AMPAR is dependent on amyloid-β as Aβ are responsible for the ubiquitination and degradation of AMPAR in primary neurons, and finally, suppressing the synaptic transmission in AD [81]. Further, the soluble form of Aβ oligomers have also been found to be involved in synaptic degradation and cognitive deficits that takes place via the subunit GluA3 AMPAR, indicating the crucial role of GluA3 in Aβ mediated degradation of synapse and cognitive functions [82]. Thus, in this scenario, it is highly crucial to implement the drugs that target the elevation of AMPARs for alleviating memory decline and cognitive impairment in AD.

## 6. Cholinergic Receptors

Cholinergic receptors are activated by acetylcholine, a neurotransmitter that is released by the motor neurons for sensory and motor processing. The cholinergic system includes nicotinic acetylcholine receptors (nAChRs) and muscarinic acetylcholine receptors (mAChRs) which are mainly responsible for signal transduction of autonomic and somatic nervous system. Nicotinic receptors function as ionotropic ligand-gated receptors and are receptive towards the agonist nicotine, whereas muscarinic receptors function as G-protein coupled receptors and are receptive towards muscarine [83]. The cholinergic neural transmission modulates memory and is associated with normal and abnormal cognitive functioning [84]. Thus, memory impairment and cognitive decline related to aging and dementia are associated with cholinergic system dysfunction [85]. Drugs acting on the cholinergic network may represent an exciting therapeutic avenue for the treatment of AD. 

AD pathology is characterized by the loss of cholinergic neurons and the gradual down regulation of the brains acetylcholine level, which is influenced by excess acetylcholinesterase (AChE) [86]. AChE, a serine enzyme found at neuromuscular joints and cholinergic synapses, hydrolyzes ACh to acetate and choline. The hydrolysis action terminates the impulse transmission at cholinergic synapses [87]. Acetylcholinesterase inhibitors (AChEi) treat AD symptoms by improving cholinergic neurotransmission; however, acetylcholine esterase inhibitors only delay cognitive decline by increasing the level of acetylcholine in the brain. It does not change AD’s underlying pathology regarding continued loss of cognitive function [88,89]. 

In the search for new medicines to treat cognition and memory, researchers continuously look for molecules and compounds which can target acetyl cholinesterase (AChE). Derived from various flavones, isoflavones, flavanols, anthocyanidins, curcuminoids and stilbenes these compounds and play an important role in inhibiting the AChE enzyme [90]. The key molecules found in the cholinergic system are shown in Figure 3. 

### 6.1. Nicotinic Acetylcholine Receptors (nAChRs)

Nicotinic acetylcholine receptors are cholinergic receptors that respond to acetylcholine and bind to nicotine. They represent the heterogeneous family of ionotropic receptors which mediate neurotransmission through ligand-gated ion channels [91]. The nicotinic AChRs are located in muscles, the central nervous system (CNS) and peripheral nervous system (PNS). They are responsible for neuromuscular transmission that causes muscular contraction at skeletal neuromuscular junction and are also involved in synaptic transmission in the CNS and PNS [92]. In addition, nAChRs also influence cognition and memory function in the brain and control the pre-synaptic release of neurotransmitters such as dopamine [93]. These receptors are large pentameric structures with a molecular mass of ~290 KD. Each receptor consists of five types of protein subunits. These subunits can either be homologous (identical subunits) or heterologous (different subunits) [94].

The nAChRs, in human beings are arranged in 16 homomeric or heteromeric subunits, consisting of a diverse set of complex subtypes such as, α1–7, α9–10, β1–4, γ, δ and ε substructures. Among these substructures, α7 and α4β2 subunits are overly expressed in the CNS. Subtype α4β2 subunits have high affinity for nicotine and cytosine and a lower affinity for α-BTX (α-bungarotoxin), whereas, α7 subunit has a high affinity for α-BTX and lower affinity for nicotine and cytosine. The subtype α7 has five subunits and is crucial in AD pathology because these subunits are involved in memory and learning functions and participates in cholinergic anti-inflammatory pathways in relation to the autoimmune disorders. The five α7 subunits are homologous receptors and are commonly referred as α7 nAChRs. It is the only α-BTX receptor identified in mammalian brain. These receptors have an N–terminal peptide and a ligand binding site that has a high affinity for α-BTX and its agonists [50]. 

Subtype α7 nicotinic AChRs in hippocampal astrocytes can regulate the calcium signaling cascade in the CNS and contribute to cholinergic signaling. These receptors exhibit a high permeability to calcium compared to the sodium ion. On activation, these receptors produce a calcium transient in the cell and increase the intracellular concentration of free calcium. The process takes place via calcium induced calcium release and is triggered by the voltage gated calcium channels. The calcium signaling cascade in astrocyte is different compared to the activation of these receptors on the neurons [95]. In neurons, the activated presynaptic α7nAChR increases the flux of Ca^2+^, depolarizes the presynaptic membrane and merges this presynaptic membrane with vesicles containing neurotransmitters in the synapse. As a result, this leads to exocytosis and induces the release of neurotransmitters, such as glutamic acid, norepinephrine (NE), ACh, dopamine (DA) and γ-amino butyric acid (GABA).

Nicotinic AChRs have important role in regulating the release of pro-inflammatory cytokines in macrophages, brain astrocytes and microglia [96]. These receptors mediate the cholinergic signaling, especially in learning and memory function. In particular, α7-nAChR modulates the excitatory neurotransmitter release, improves learning and memory ability and enhances the cognitive function. It has been reported that in the brain of AD animal models and AD patients, the functioning level of α7nAChRs along with the level of expression is altered [97]. Higher level of α7nAChR is observed in the brain of early embryonic stage and this level progressively changes with age, inferring the importance of α7nAChR in growth, development, and aging [87]. Thus, α7nAChR may participate in AD pathogenesis and may serve as a novel therapeutic target for AD treatment.

#### 6.1.1. Interaction between Amyloid Beta and α7nAChRs 

The subunit α7nAChR is involved in cholinergic signaling and has been associated with amyloid beta deposition and AD pathogenesis. Relevant to cognition, synaptic plasticity and the neurotoxic properties of Aβ peptides, several studies have proposed both agonistic and antagonistic relationships between Aβ and α7nAChRs. Activation or inhibition of α7nAChRs blocks Aβ peptide mediated neuronal cell death and is considered a critical step in the identification of potential therapeutic strategies to treat AD pathogenesis. Accumulated evidence suggests that subtype α7 nicotinic acetylcholine receptors mediate Aβ induced neurotoxicity in hippocampal neurons [98] and Aβ induced tau phosphorylation via activation of tau kinases, extracellular-signal-regulated kinase (ERK), and c-Jun N-terminal kinase (JNK-1) [99]. In these cases, α7nAChRs demonstrates an agonistic effect on Aβ peptide induced neuronal cell death. Inhibition of the α7nAChR subunit, rather than activation, may imply a pathway that eliminates the neurotoxic properties from Aβ peptides. Aβ induced neurotoxicity is mediated by the up regulation of α7nAChRs in hippocampal neurons. Liu et al. emphasized that α7nAChRs up regulation produces Aβ induced neuronal hyperexcitation and possibly, AD pathogenesis [100]. The up regulated α7nAChRs are also responsible for the down regulation of ERK2 mitogen-activated protein kinase (MAPK) activity that is critical in hippocampus synaptic plasticity and learning. The in vitro and in vivo studies demonstrate that Aβ42 pairs up with MAPK cascade via α7nAChRs, leading to the up regulation of α7nAChR. The ERK MAPK cascade may regulate the production of Aβ42 peptides and the down regulated ERK2 MAPK may alleviate the accumulation of Aβ42 peptides [101].

Immunohistochemical studies of human brain tissues confirm the agonistic relationship between Aβ peptides and α7nAChRs. Wang et al., illustrated that Aβ 1-42 has high affinity with α7nAChR and binds to form a stable complex that inhibits the release of ACh and alters Ca^2+^ homeostasis in the cholinergic neuron causing neuronal dysfunction (Figure 4). They have also reported that Aβ1-42 mediates the death of human neuroblastoma cells that overexpress α7 nicotinic acetylcholine receptors, which contrast with α7nAChR agonists, nicotine and epibatidine that protect human neuroblastoma cells from neuronal death induced by amyloid beta peptides [102]. Likewise, in vitro studies have also shown that nicotine inhibits the formation of Aβ fibril from Aβ1-42 and disrupts preformed Aβ fibrils protecting the neurons from Aβ toxicity via the up regulation of nicotinic receptors [103]. Similar studies have also provided evidence demonstrating nicotine’s effectiveness at reducing the insoluble Aβ peptide plaques in treated transgenic mouse brain of [104]. Together, these studies suggest that the reduction of amyloid beta peptide in brain may be mediated by α7nAChRs and nicotinic drug may prove a novel protective AD therapy.

The cerebral cortex of AD patients has low AChE levels within the amyloid plaques [105]. The activity of AChE increases around the amyloid plaques and is believed to be caused by the direct action of Aβ on AChE [106]. AChE is a cholinergic enzyme that terminates the synaptic transmission between the synapses [107]. Fodero et al. have demonstrated that the effect of Aβ1-42 on AChE is due to the agonist effect of Aβ on α7nAChRs. In primary cortical neurons, α7nAChRs mediate the Aβ1-42 induced AChE increase. The inhibitors of α7nAChRs and L- or N- type voltage dependent calcium channels (VDCCs) block the effect of Aβ on AChE and α7nAChR agonist s increases the level of AChE [108]. In this case, AChE inhibitors have been used to reduce ACh breakdown and enhance their levels. This subsequently makes the neurotransmitter available to the nicotinic and muscarinic receptors, enabling them to increase cholinergic signaling and thereby reducing AD memory deficits [109].

AD is pathologically characterized by the presence of extracellular amyloid plaques and intracellular NFTs. The α7nAChR subtype is involved in synaptic plasticity in the brain and Aβ1-42 is closely associated with AD pathogenesis. Such agonistic results suggest that Aβ may exert some of its toxicity through α7nAChRs and thus provide a possible pathway for AD treatment by blocking the action of Aβ 42 on α7nAChRs. In contrast, several other reports have elucidated the antagonistic relationship between α7nAChRs and Aβ peptide. The nicotinic acetylcholine receptors with the subunit α7 mediate synaptic current and plasticity in the brain. They modulate cellular function in the nervous system and their responses affect cognitive processes and memory forming abilities. However, the responses of α7 containing nicotinic receptors are blocked by the presence of nanomolar concentrations of Aβ1-42 in the hippocampal neurons, thereby affecting their proposed cognitive roles and leading to learning and memory dysfunction [110]. In a different study, Puzzo et al. demonstrated that a picomolar concentration of Aβ, exerted a positive modulatory effect on hippocampal synaptic plasticity and memory, whereas a higher concentration resulted in neuronal dysfunction and cognitive failure [111]. Nonetheless, these studies suggest that nicotinic receptors are a subject of interest for treating AD pathogenesis. Their central role in synaptic regulation offers an exciting opportunity for therapeutic development that specifically targets these receptors or alternatively intercepts the action of Aβ peptides on the nicotinic receptors.

#### 6.1.2. Allosteric Modulation of nAChRs

In addition to the hallmark Aβ plaques and hyper phosphorylated tau proteins, neuroinflammation appears to have a significant role in the development and progression of AD pathogenesis. The neuro-inflammatory process is marked by microglial activation and is considered to be induced by the binding of Aβ peptides to cluster of differentiation 36 (CD36) protein and toll like receptors (TLR) of four and six heterodimers, such as TLR4 and TLR6 [112]. Activation eventually leads to the production of proinflammatory cytokines, for example, interleukins (IL-1β, IL-6, IL-8) and tumor necrosis factor (TNF), along with anti-inflammatory cytokines, such as transforming growth factor β (TGFβ), chemokine, and small messenger molecules, namely, nitric oxide and reactive oxygen species, which over time, cause nerve impairment and neuronal death [113]. 

In this context, the microglial activation leading to neuroinflammation has also been reported to take place via the brain cholinergic pathways involving α7 nAChRs, where the release of microglial TNF-α, as induced by lipopolysaccharide (LPS) through activation of α7 nAChRs, is regulated by the acetylcholine and nicotine levels [114]. The use of nAChR allosteric modulators is a promising strategy to reduce neuroinflammation [65] earlier. The current treatment, which uses nAChR agonist and AChE inhibitors, is limited by their desensitization properties [115].

In AD, allosteric modulation intensifies the ACh activity at pre- and post-synaptic nAChR. This is particularly important because the presynaptic ACh level is significantly lower and impaired in AD patients. Allosteric modulators of nAChRs increases the presynaptic ACh level and enhances the cholinergic nicotinic neurotransmission by amplifying the interaction between the nAChR and ACh. This presynaptic release of ACh is considered to be mediated by α7 and α4β2 nAChR [116]. Accordingly, some of the allosteric modulators of nAChRs are capable of producing positive modulatory effects on nAChR. These positive allosteric modulators (PAMs) may have therapeutic potential. For instance, desformylfulstrabromide (dFBr), PAM of α4β2 receptors, enhances cognition in rat [117]. Galantamine, type I PAM for α7 and α4β2 nAChR, elevates ACh levels and ameliorates cognition in AD patients [118], and cotinine, PAM of α7 receptor, is neuroprotective, improves memory in primates, and lowers the Aβ burden in AD mice [119]. 

### 6.2. Muscarinic Acetylcholine Receptors

The cholinergic receptors most responsive to muscarine are referred to as the muscarinic acetylcholine receptors (mAChRs). Members of the metabotropic receptor class, mAChRs mediate the physiological action of acetylcholine in the CNS and PNS using G-protein for as the signaling mechanism [120]. Found throughout the human body (e.g., renal and cardiovascular systems, gastrointestinal tract, eyes, brain, salivary glands) mAChRs are responsible for many distinct physiological functions including motor control, cognition, and sensory processing, dependent upon location and receptor subtype [121,122]. The mAChRs are the receptive group of the acetylcholine neurotransmitter and mediate cholinergic transmission related to learning, memory, and cognition in the forebrain region thus the implication in AD [123]. 

Muscarinic receptors are part of the ligand-gated G-protein coupled receptor and perform either as a simulative regulative G-protein (Gs) or an inhibitory regulative G-protein (Gi) [124]. The muscarinic AChRs have been divided into five receptor subtypes, M1-M5. Subtypes M1, M2 and M4 muscarinic receptors are adequately present in the brain, whereas M3 and M5, are sparsely distributed [125,126]. The receptors M1, M3, and M5 are stimulatory receptors that interact with the Gq/11 G protein family and are involved in the stimulation of phospholipase C and instigation of the phosphatidylinositol trisphosphate cascade, resulting in the intracellular Ca^2+^ mobilization and the activation of protein kinase C. In contrast, the inhibitory receptors M2 and M4, interact with the Go/i family of G proteins. This set of receptors are responsible for inhibiting adenylyl cyclase i and activating G protein-gated potassium channels. The inhibitory action of adenylyl cyclase reduces the level of protein kinase A, ultimately decreasing cAMP (cyclic adenosine monophosphate) levels in the cell [127]. 

The subtype M1 muscarinic receptor extends throughout the brain and is found primarily in the hippocampus, cerebral cortex, amygdala, and corpus striatum [127]. Amyloid plaques and NFTs develop in the cerebral cortex and hippocampus Given that M1 receptors are highly expressed in these sections, the M1 muscarinic receptor is a potential therapeutic target for restoring cholinergic signaling [128]. M1 muscarinic receptors are associated with numerous physiological functions, including neuron excitation, synaptic plasticity, learning and hippocampal-based memory, neuronal differentiation during early development, modulation of cognition, and short-term memory [129]. Besides the obvious AD associated plaques and tangles, presynaptic cholinergic hypo function is also considered an equally attributive factor in progressive dysfunctional cognition. Lending weight to the cholinergic hypothesis, post-synaptic M1 mAChR remains unaffected, and in this case, several studies have suggested M1 muscarinic agonists as prospective AD treatment candidates [130].

M1 muscarinic agonists have been reported to reduce amyloid-β peptides and tau pathologies in the hippocampus and cortex regions of an AD mouse model. Caccamo et al. administered AF267B, a low molecular weight M1 muscarinic agonistic, in the 3xTg-AD model of Alzheimer disease. 3xTg-AD model mice experienced a reduction in both Aβ and tangles in the hippocampus and cortex. Cognitive deficits related to spatial tasks were additionally reversed. Likewise, they also reported that dicyclomine, M1 antagonist, amplified Aβ and tau pathologies, highlighting the importance of M1 agonist against the biomarkers of AD pathologies. M1 activators modulated the amyloid precursor protein (APP) processing away from the amyloidogenic pathway by generating α-secretase products via the activation of protein kinase C (PKC) and extracellular regulated kinase (ERK ½), ultimately diminishing the Aβ peptides. The activation of PKC also reduced the activity of tau kinase, glycogen synthase kinase (GSK) 3β, decreasing tau pathologies in the AD animal model [128]. The M1-selective agonist, VU0364572, was administered to 5XFAD mice, by another group, which showed that levels of soluble and insoluble Aβ40, 42 were significantly reduced in the hippocampus and cortex of the animal preventing memory impairment as demonstrated by the Morris water maze task [131]. These findings reinforce the development of M1 activators as a disease-modifying treatment for AD pathology. The activation of PKC is linked with increased APP metabolism, secretion, and processing [132]; selective cholinergic agonists and M1 muscarinic agonists increase PKC activity and serve as an effective AD modifying therapy [133]. M1 mAChR’s significance to APP processing and amyloid pathology has also been demonstrated in vitro and in vivo AD transgenic mouse model studies. Removal of the receptor increased amyloid pathology, indicating that M1 mAChR is instrumental in modulating amyloidogenic APP processing in neurons with M1 activators as potential therapeutic resources for treating AD [134]. 

Like M1 agonists, M2 antagonists are also considered a prospective therapeutic measure for treating AD behavioral and cognitive symptoms. As mentioned previously, increased M2 mAChR activity decreases the adenylyl cyclase activity, which later decreases cAMP levels, ultimately leading to memory loss as well as reduce synaptic plasticity. Decreased cAMP levels correlate with depression and dementia severity. It has been reported that blocking M2 mAChR increases the level of ACh in the brain. The elevation of ACh then activates the postsynaptic M1 mAChR, which is involved in cholinergic signaling, eventually improving the cognitive processes [135]. 

Acetylcholinesterase inhibitors (AChE-Is), such as donepezil, galantamine, tacrine, and rivastigmine have been used for the symptomatic treatment of AD to ameliorate the cognitive function status. These inhibitors do not modify the disease progression, but rather they inhibit the action of the AChE enzyme on ACh and increase the ACh level to bind with AChRs to improve cholinergic signaling [136]. However, this treatment lasts for shorter period of time and is not considered effective. M1 muscarinic agonists offer a greater advantage than AChE-Is and can activate cholinergic receptors, which is less susceptible to ACh degeneration of. The successful development of M1 selective agonists is essential to treating AD pathology. 

Various studies have demonstrated that muscarinic and nicotinic agonists are viable therapeutic targets for the treatment of AD. Several preclinical and clinical phases studies have shown that the agonists viz. xanomeline, encenicline, nelonicline (ABT-126), AF102B effectively treat mild to moderate stages of AD through neural regeneration and decreasing amyloid-β concentration [137]. Most of these agonists mimic the action of the physiological neurotransmitter (ACh) found in the CNS. Although these agonists are possible therapeutic targets, and passed initial clinical trials, they were suspended or terminated during phase three of their respective clinical trials. Reasons for failure included cross-activity with other receptors, especially 5-HT3 and insufficient selectivity with cholinergic receptors. Rather than developing a thorough understanding of AD mechanisms and recruiting diverse populations for clinical studies, most of the agonists were designed solely to decrease amyloid-β concentration Most of the agonists were designed solely to decrease. Many laboratories remain focused on cholinergic receptor agonists by manipulating or modifying existing drugs or by designing completely new moiety using computational drug design tools. Enrolling patients during the earliest stages of the disease, selecting more diverse participants, and paying attention to specific structure-based and ligand-based drug design could increase AD treatment therapeutic target success. 

## 7. Gamma Amino Butyric Acid Receptors 

γ-Aminobutyric acid (GABA) is a predominant neurotransmitter in the mammalian central nervous system that inhibits or blocks certain brain signals and decreases neuronal activity in the nervous system [138]. This amino acid plays an important role in neuronal function and homeostasis, mainly in the hippocampus and neo cortex, the region in the brain associated with spatial memory and synaptic plasticity. A potential target for AD pathogenisis, GABAergic signaling is responsible not only for AD but also contributes to a number of neurological disorders (e.g., anxiety, autism, depressive and bipolar disorders, schizophrenia) by contributing to the regulation of cognition, learning and memory, neural development, motor function, adult neurogenesis and sexual maturation [139].

GABA is an amino acid synthesized at pre-synaptic terminals by the action of glutamic acid decarboxylase (GAD) on the glutamate molecule. It is then packed into the synaptic vesicles by vesicular GABA transporter (vGAT) and then released into the synapse or synaptic cleft by membrane depolarization (Figure 5). The inhibitory action of GABA is expressed through two different types of receptors, namely GABA_A_ and GABA_B_. GABA_A_ is an ionotropic receptor for the chloride ion channel and GABA_B_ is a metabotropic receptor that modulates the ion channels via G-protein [140]. 

The GABA receptor with subtype A, GABA_A_, is a ligand gated ionic channel permeable to Cl^-^ and HCO^3-^ anions, that integrally forms a Cl^-^ complex channel characterized by a rapid inhibitory (within milliseconds) synaptic transmission in the basal ganglia network. These receptors are heteropentameric structures formed by the combination of five different subunits, coming from a family of 19 subunit assemblies with distinct genes (α1-α6, β1-β3, γ1-γ3, δ, ∈, θ, π and ρ1-ρ3). The isomeric form made by arranging two α1-subunit, two β2- subunit, and one γ2- subunit is the most predominant form in the receptor [141]. Since the chloride channel is an integral part of the GABA_A_ receptors, the GABA binding site is directly involved in opening the of Cl^-^ channels. And as such, these receptors comprise five different key binding sites that are localized in or near the chloride channel for GABA, barbiturates, benzodiazepines, anesthetic steroids and picrotoxin, making them a critical molecular target for countless drugs. Furthermore, these receptors are modulated by neuroactive steroids, enhancing the function of GABA_A_ receptors in the brain [142]. 

In contrast, GABA_B_ receptors are GPCRs that convey a slow, steady, and prolonged inhibitory effect through the G-proteins. The inhibitory action can take place in both the presynaptic and postsynaptic neuron terminals. GABA_B_ receptors are heterodimers of R1 and R2 subunits, which are instrumental for mediating the GABA actions in relation to the GPCR function. There are three binding sites in these receptors, namely the Venus flytrap domain (VFT, an extracellular N-terminal site), an intracellular C-terminal domain, and finally the heptameric transmembrane protein [143]. GABA_B_ receptors are indirectly associated with K^+^ ion channels for increased the membrane potential. On activation, these B subtype GABA receptors reduce the calcium ion channels, inhibiting the adenylate cyclase and intracellular production of cAMP [144]. 

## Relation between Amyloid Beta, Tau Formation and GABAs

The aggregation of amyloid beta peptide and microtubule related protein tau are the biomarkers of AD pathology. Amyloid precursor protein (APP) is the parent polypeptide for the generation of Aβ peptide by the action of β- and γ-secretases on APP. In the mammalian brain, APP is involved in regulating synaptic functions and neuronal survival. They are directly associated with GABAergic signaling where they modulate potassium chloride co-transporter 2 (KCC2) and the activity of presynaptic GABA_B_ receptors, suggesting that APP and its proteolytic compound influences the development and functioning of the mammalian CNS [145]. GABA_B_ receptors can form complexes with APP, leading to presynaptic axonal trafficking, which ultimately is responsible for the formation of Aβ peptides. Since the malfunctioning axonal trafficking and down regulation of GBR is responsible for formation of amyloid beta, securing APP with GB1a at the surface of the cell can prevent the formation of Aβ. The complex formation of APP and GABA_B_ receptors stabilize APP, thereby restricting the accessibility of APP for amyloidogenic processing, and may provide AD symptomatic improvement [146].

The expression of the GABA_A_ receptor’s α6 subunit and the maturation of the cerebellar granule neuron (CGN) in rat are modulated by Aβ40 peptides through the extracellular-signal-regulated kinase/mechanistic target of rapamycin (ERK/mTOR) pathways. It is believed that CGNs in the presence of Aβ40 significantly induce the phosphorylation of ERK and mTOR, eventually increasing the expression of the GABA_A_ α6 subunit, affecting the development of CGN in the rat models [147].

Another hallmark of AD pathology, the formation of intracellular neurofibrillary tangles or NFTS, results from the hyper phosphorylation and aggregation of the tau protein thought to be modulated by the activated GABA receptor, which induces phosphorylation at the AT8 epitope (Ser-199/Ser-202/Thr-205) in cultured mature cortical neurons. As a result, increased phosphorylated tau levels reduce tau proteins with protein phosphatase 2A (PP2A), a serine/threonine protein phosphatase. The PP2A activity is related to neurodegeneration and has a critical role in signal transduction and cell transformation. The decreased protein-protein interaction between the tau and PP2A, in this case, depends on cyclin-dependent kinase 5 (Cdk5), despite the absence of Cdk5 glycogen synthase kinase-3β (GSK3β) activity [148]. It has also been proposed that GABA_A_ receptors are involved in anesthesia-induced hyper phosphorylation of tau proteins, which eventually leads to the development of NFTs and neurodegeneration. The binding of GABA_A_ receptors with anesthetics may be responsible for inducing tau phosphorylation, that when activated by GABA_A_ receptors, enhances the PP2A level. Enhanced PP2A may be responsible for the dephosphorylation of beta subunits rather than the dephosphorylation of tau proteins. Increased tau protein phosphorylation processes may ultimately lead to diminished memory function and impaired synaptic plasticity [149].

GABA is the main inhibitory neurotransmitter in the mammalian CNS and is regarded as a promising therapeutic target for AD pathology. Generally considered postsynaptic receptors, GABA receptors are characterized as extra synaptic receptors mediating tonic conductance in the cerebellum granule neurons (CGNs), dentate gyrus (DG), and hippocampal interneurons [150]. Reducing the inhibitory action of GABA on tonic conductance in the DG may serve as a novel therapeutic approach for AD pathology. Aβ peptide and tau proteins are AD general biomarkers; however, recent studies reported elevated GABA in the astrocyte’s DG, suggesting that GABA upregulation may result in enhanced tonic inhibition, which subsequently may cause long term potentiation and memory impairment. Interestingly, in 5XFAD mice, reduced tonic inhibition rescued LTP impairment and memory dysfunction, offering another therapeutic target for AD pathogenesis [151].

GABA receptors have the potential to treat both early and later stages of AD pathogenesis but have received relatively little attention as a therapeutic target. Few drug candidates, including antagonists and allosteric modulators SGS742 and EHT0202 (etazolate hydrochloride) andβ-carbolines have successfully completed preclinical and clinical trials. Intended to treat mild to moderate AD, the drugs require further evaluation in large multi-ethnic populations at different stages of the disease [152]. In depth knowledge of the GABA receptor’s subunit expression and complex regulation will further advance its viability as a potential AD drug target. 

## 8. 5-Hydroxytryptamine Receptors

5-Hydroxytryptamine (5-HT) receptors, also referred to as serotonin receptors, are a monoamine receptor molecule, activated by the neurotransmitter serotonin. These receptors are responsible for mediating both the inhibitory and excitatory synaptic transmission in CNS and PNS. And, depending on the nature and type of 5-HT receptors, they can function either as a G-protein coupled receptor or as a ligand gated ion channel receptor [153]. Considered one of the largest groups of receptors found in mammal, 5-HT receptors can be categorized into a family of six GPCRs, namely 5-HT1, 5-HT2, 5-HT4, 5-HT5, 5-HT6 and 5-HT7, and one ionotropic receptor, 5-HT3. Combined, 5-HT consists of 14 distinct serotonin receptor subtypes with distinguished structure and pharmacological activities [154]. Except for 5-HT3, an ionotropic receptor, the seven membrane GPCR-type of serotonin receptors can couple with G-proteins, for instance, Gαq/11, Gαi/o, or Gαs, to produce a nerve signal, and as such can be classified on the basis of their signaling mechanism. The 5-HT3 receptor is a ligand gated ion channel receptor that is permeable to cations such as Na^+^, K^+^ and Ca^2+^ for modulating the excitatory neurotransmission in CNS and PNS [155].

5-HT1 receptors with subtypes 5-HT1A, 5-HT1B, 5-HT1D, 5-HT1E, and 5-HT1F, and 5-HT5 receptors with subtypes 5-HT5A and 5-HT5B, bind with Gαi/o – protein to mediate neurotransmission, such as adenylyl cyclase inhibition and decreased cAMP levels [156]. 5-HT2 receptors, comprise of subunits 5-HT2A, 5-HT2B, and 5-HT2C, bind with Gαq/11, a protein that initiates the activation of phospholipase C (PLC) for hydrolyzing phosphatidylinositol bisphosphate with the generation of inositol trisphosphate 3 (IP3), mobilization of intracellular Ca^2+^, influx of extracellular Ca^2+^ and activation of protein kinase C (PKC) [157]. Among the 5-HT2 receptor subtypes, 5-HT2A is well distributed in the CNS, primarily in the brain region responsible for cognitive and learning functions making them a novel target for therapeutic development to address AD and other neurodegenerative diseases [158].

Similarly, the remaining GPCR- type receptors, 5-HT4, 5-HT6, and 5-HT7 bind with Gαs-protein to mediate CNS excitatory neurotransmission. 5-HT4 and 5-HT7 are widely expressed in the central and peripheral CNS and are involved in modulating gastrointestinal functions and enhancing the cAMP production level [159]. Likewise, 5-HT6 is also highly abundant in nucleus accumbens, striatum, cerebral cortex, hippocampus and olfactory tubercle, and is involved in enhancing cAMP signaling and stimulating the adenylyl cycle. These receptors also have a key role in cognition and memory function, where the agonists and antagonists of 5-HT6 receptors are believed to ameliorate glutamatergic and cholinergic mediated learning and memory impairments [160]. 5-HT receptors and their subtypes, particularly due to their ability to modulate serotonin receptors by agonists and antagonists, provide a promising drug target to mitigate denervation in AD pathology.

Considered one of the oldest neurotransmission systems in the animal kingdom, serotonin plays a pivotal part in cognition and behavioral control and is implicated in AD pathology [161]. Post-mortem studies conducted on AD brains verify the direct involvement of the serotonergic system in denervation. Serotonergic processes are considered dysregulated in AD patients including subsequent alterations in 5-HT system function that leads to depression, one of the many behavioral changes experienced by AD patients [162]. The administration of selective serotonin reuptake inhibitors (SSRI) have been used as a long-term anti-depressant tool to slow the advancement of AD. In the mouse overexpressing APP/PS1 model of AD, SSRIs reduced the level of Aβ found in the brain’s interstitial fluid, particularly the ERK pathways, confirming that ERK is necessary for the serotonin-dependent depression of ISF Aβ. SSRIs are considered safe neuroactive compounds with minor side effects. Clinical studies additionally determined that SSRI antidepressant drug treatment significantly reduced the amyloid load in cognitively normal elderly patients as compared to those who were not cognitively normal. These studies suggest that serotonin signaling is linked to decreased amyloid and plaque reduction in both transgenic mice and humans [163]. Several studies have been carried out to identify the specific 5-HT receptors that are involved in modulating the serotonergic signaling for AD treatment.

5-HT4 receptors mediate serotonergic induced memory and learning functions in the hippocampus, cortex, and striatum [164]. On activation, the agonists BIMU 1 and BIMU 2 facilitate the release of acetylcholine. Thought integral to cognition, experiments conducted on rat frontal cortex, strongly suggest that serotonergic activity is pivotal to improving cholinergic impairments related to memory and learning dysfunction [165]. Besides the cholinergic system, several studies have also concluded that 5-HT4 receptors and their agonists modulate the non-amyloidogenic processing of APP responsible for generating the soluble amyloid beta precursor protein (sAPPα). sAPPα, considered a neuroprotective protein, reportedly increases the amount of these soluble fragments and is another likely approach to treat AD. 5-HT4 receptor agonists have been shown to increase sAPPα levels in the cortex and hippocampus areas of the mouse model, making them a promising pharmacological target for AD treatment [36,166]. In the non-amyloidogenic APP metabolism, 5-HT4 receptors, in the absence of agonist activation, are believed to enhance the production of sAPPα fragments by interacting with ADAM10 with the subsequent release of the soluble APP proteins in HEK-293 cells and cortical neurons. This process is independent of cAMP production, however, in the presence of an agonist, the 5-HT4 receptors are triggered to enhance the formation of sAPPα proteins via cAMP/Epac signaling, conferring the role of 5-HT4 receptors in α-secretase ADAM10 and sAPPα release [167]. Similarly, the potentiality of type 4-hydroxytryptamine receptor as an AD modifying agent has been supported by the chronic administration of 5-HT4 receptor agonist RS67333 during the prodromal phase of the disease. In this case, the agonist compound was reported to reduce the Aβ levels in 5XFAD mouse models with successive improvement in NOR (novel object recognition) test impairments 168. In addition, SL65.0155, an activator of 5-HT4 receptors, has also been reported to ameliorate learning and memory performances in object recognition task by activating cAMP production [168,169].

Serotonin-6 receptors (5-HT6) are similar to the type 4 (5-HT4) receptors and have also garnered considerable attention as a promising candidate pharmacological intervention in AD pathology. Several studies have introduced a myriad of antagonists, to ameliorate serotonergic mediated cognitive dysfunction and memory performance in a number of behavioral tests conducted on memory deficient adult male Wistar rats. For instance, an activator of serotonin-6 receptor, Ro4368554, has been shown to enhance the serotonergic and cholinergic cognitive effects in object recognition tasks [170]. Similarly, dimebolin, an antagonist of 5-HT6 receptors is thought to boost cognitive function [171]. Idalopirdine (Lu AE58054), a selective 5-HT6 receptor antagonist, combined with donepezil, was shown to ameliorate the cognitive functions in mild AD cases [172]. Another compound, SB271036, improves memory performance by reducing Aβ generation. SB271036 inhibits the activity of γ-secretase and inactivates astrocytes and microglia in the mouse model of AD [173].

5-HT4 receptor activators and 5-HT6 receptor inhibitors of receptors have been considered excellent drug targets in several clinical trials, due particularly to their memory and behavioral performance. Both of these receptors have amassed significant attention from the AD research community as promising therapeutic targets compared to the other receptors. Nonetheless, a number of studies have also noted the potential promise of 5-HT1 receptors in terms of memory and behavioral. 5-HT1A receptor antagonists are believed to activate glutamatergic and cholinergic neuro signal and subsequently enhance cognitive impairments as observed in AD pathogenesis [174]. Postmortem AD tissue correlates with known aggressive patient behavior providing the basis for developing 5-HT1A receptors as a target for treating behavioral symptoms in AD pathology [175]. Likewise, administration of S15535, agonist of 5-HT1A receptor, increased the response accuracy and reduced the delayed response in mouse models. In addition, these receptors increased the release of ACh in the frontal cortex and dorsal hippocampus regions of freely moving rat model, thereby facilitating cognitive function in a varied behavioral performance [176]. In a recent study, 5-HT1A receptor inhibitor (NAD-299) and 5-HT2A receptor activator (TCB-2) have been shown to decrease neuronal loss and oxidative stress in a rat model of AD, implying that these receptors may also be explored as a preventative for AD progression [177]. 

Serotonergic neurotransmissions are an integral system that modulate hippocampal and neo cortical cognitive and learning performance. This system, by its very nature, leads to denervation and age-related cognitive and behavioral disorders, thereby serving as a potential therapeutic target in AD pathology. Consequently, AD progression can be ameliorated by limiting serotonin receptor functioning through the use of agonists for 5-HT4, 5-HT2A/2C and antagonists of 5-HT6, 5-HT1A or 5-HT3 and 5-HT1B [178]. The use of either the activator or inhibitor of a specific serotonin receptor not only prevents memory impairment, but also enables learning processes in situations requiring high cognitive demand. Together, this provides a novel therapeutic opportunity for treating AD.

## 9. Amylin Receptors

Amylin, a peptide hormone with 37 residue units, has been linked as a putative target for cognitive damage and glycemic irregularities in relation to Type 2 diabetes mellitus (T2DM) and obesity [179]. Amylin, or islet amyloid polypeptide (IAPP), is secreted along with insulin by the pancreatic β-cells [180]. These pancreatic polypeptides form a class of calcitonin (CT) peptide family and share a similar structure with CT, adrenomedullin, and CT gene-related peptide (CGRP). Receptors whose binding unit comprises CT and RAMP (receptor activity modifying protein) form the base of amylin receptors (AmRs). AmRs are a heterodimer made by combining CTR subunit (calcitonin receptor) and one of the RAMP members (RAMP1, RAMP2, or RAMP3)., Three types of amylin receptors have been identified, namely AMY_1_, AMY_2_, and AMY_3_. CTR is a type of GPCR, and its binding affinity with amylin can be modified in the presence of RAMPs [181]. In this scenario, the hetero structures of either the CTR and RAMP1 or CTR and RAMP3 have been considered to favorably bind with amylin to mediate physiological functions in the brain. Both CTR and RAMPs manifest in the brain including hippocampus, cortex, and locus coeruleus, the regions most associated with AD [182]. 

The amylin polypeptides that accumulate within T2DM pancreatic islets have similar physiological properties with amyloid beta peptides deposited in the brains of AD patients. Both diseases share pathogenic similarities including amyloid aggregation, inflammation, oxidative stress and neurotoxicity [183]. At cellular level, both Aβ peptide and IAPP (islet amyloid polypeptide) uses amylin receptor to express their biophysical effects leading to cytotoxicity. The process of cytotoxicity is achieved by activating GPCR and inducing the common intracellular pathways including, the signal transduction mediators such as protein kinase A, MAPK, protein kinase B, and cFos. Thus, AMY3 receptor (amylin-3 receptor) may serve as a therapeutic target for the action of Aβ peptide for the treatment of AD [184,185]. The effect of Aβ appears to take place via the amylin receptor in the cell, showing direct interaction between Aβ oligomer and amylin receptors. Indeed, the action of the two peptides can regulate the neuronal activity at cellular and synaptic level by activating the amylin receptor. However, the prolonged exposure of the receptor to the peptides results in apoptosis via the signal transduction pathways (Figure 6). 

Other approaches have suggested the beneficial role of amylin receptors antagonists as a promising therapeutic agent in AD. The depressant effects of Aβ (1-42) and human amylin on hippocampal long-term potentiation (LTP) were blocked by the application of AC253, an antagonist developed for the treatment of T2DM, in the APP transgenic mouse model [186]. Likewise, the amylin receptor antagonist, AC253, has also been reported to block the electrophysiological effects of Aβ on human fetal neurons (HFNs). It has also been observed that expression of amylin receptors is correlated with amyloid burden. The down regulation or blocking of amylin receptor using siRNA in HFNs is considered to provide protection to the neurons from Aβ induced apoptotic cell death [187].

In humans, lower concentration of the pancreatic hormone amylin has been connected with cognitive impairment [188]. The administration of increased levels of amylin in the murine mice model of AD reduced the Aβ in the brain with significant behavioral improvement [189]. Similarly, in another study APP transgenic mice received an intraperitoneal injection of amylin and pramlintide, an analog of amylin, successfully reducing the level of Aβ by removing the peptide from the brain. Further, in Morris water maze and Y maze tests, the animal showed remarkable improvement in learning and memory functions [190]. Amylin’s utility is not limited to minimizing the Aβ peptide in the brain but has also been shown to diminish tauopathy and brain inflammation in two animal models of AD. Subsequent administration of human amylin remarkably decreased cerebral Aβ and reduced tau proteins, insoluble tau, and Iba1 and CD68 (inflammatory markers). Amylin and its receptors, in this case, are thought to influence the CDK5 signaling by decreasing the active form of CDK5, p25, by reducing the tau phosphorylation [191]. Indicating, that amylin can also act as a potential therapeutic approach for ameliorating neurodegeneration and cognitive defects in AD. 

Amylin’s role remains ambiguous. Some studies portray it as a critical component to reduce amyloid beta peptides and restore cognitive functions. Other studies consider it a causative factor for neurotoxicity and long-term potentiation in the hippocampus. Lim et. al. proposed that the action of human amylin, unlike rat amylin, has strikingly similar neurotoxic features to those shown by Aβ peptides. In this study, they proposed that both the human amylin (hamylin) and amyloid-β peptides were toxic to hippocampal and cortical neurons [192]. The toxic effect of hamylin in rat hippocampal neurons were reported to be concentration dependent and thought to occur via transient receptor potential cation channel subfamily V member 4 (TRPV4) channels. At high concentration of human amylin, these polypeptides activated the TRPV4 channel, which subsequently depolarized the local membrane and allowed a high calcium influx into the cells resulting in neuronal dysfunction, inflammation, and neurotoxicity, thereby making TRPV4 a therapeutic interest for treating AD pathology [193]. Evidence from the studies indicate that amylin’s role in neurodegeneration should not be neglected. The amylin analogue, pramlintide, along with the amylin receptors and their antagonists are significant factors that define AD’s etiology, reduce the amyloid plaques in AD brain, and subsequently restore cognitive impairments.

Amylin receptor antagonists could be potential AD drug targets due to its presence in central nervous system cells and is directly associated with amyloid beta formation upon activation. In addition, no evidence supports the direct connection of amylin receptor function with other receptors such as dopamine and serotonin for its regulatory function.

## 10. Netrin Receptors

Netrin–1 are chemotropic proteins that belong to the family of laminin-related secreted proteins [194]. Involved in axon guidance, these proteins are located in the somatic mesoderm of the central nervous system and in the ventral region of the spinal cord, specifically in the neuroepithelial cells and the floor plate, and non-neuronal areas such as the pancreas and cardiac muscle [195]. These proteins are responsible for guiding cell and axon migration during neural growth and development [196]. They possess antiapoptotic [197] and anti-inflammatory properties [198], and are instrumental during neurogenesis, angiogenesis, and morphogenesis [199]. netrin–1 response is mediated by the family of UNC5 and DCC (deleted in colorectal carcinoma) receptors. Netrin–1 is bi-functional in nature, attracting some axons while repelling others [200]. UNC5 is responsible for directing the signal away from the netrin source (chemo repulsion), while, DCC is responsible for chemo attraction, but only when located at a distance from the source of netrin–1. DCCis also involved in repulsive processes [201].

Netrin–1 receptor has proapoptotic activity. Thus, in the absence of netrin-1 receptor, DCC and UNC5, netrin dependent receptors, incite cellular apoptosis [202]. Where, UNC5 receptors mediated apoptosis by activating death-associated protein kinase (DAPK), particularly, the abnormal UNC5C contributes to AD by activating death-associated protein kinase 1 (DAPK1) [203]. Furthermore, the deficiency of netrin–1 has also been shown to cause a missense mutation (T835M) in the UNC5C receptor, resulting in late-onset Alzheimer’s disease (LOAD). The over expression of T835M-UNC5C can induce the nerve cell death and increase the risk of LOAD, however, the neuronal cell death can be inhibited by netrin-1 binding to UNC5C on the cell surface [204]. Nonetheless, the molecular mechanism of the receptor is unclear. Recently, Chen et. al. have shown that neurodegeneration in AD is facilitated by the selective cleavage of UNC5C, by the action of δ-secretase, contributing to AD pathogenesis. δ-secretase is activated in the absence of netrin which then distinctively cuts UNC5C at the N467 and N547 residues, supplementing caspase-3-activation and neuronal cell death. In APP/PS1 mice, the expression of δ-secretase truncated UNC5C fragments enhanced AD pathologies, thereby reducing learning and memory capabilities. Simultaneously, when UNC5C was removed from these mice, they regained cognitive disorder and weakened AD pathogenesis [205].

In addition, gene polymorphism in netrin–1 and its receptor have also been linked with AD pathogenesis. In the adult mammalian brain, mutations in the receptor proteins are involved in neurodevelopment and neurodegeneration disorders. Synaptic plasticity during learning and memory development is controlled by α-amino-3-hydroxy-5-methyl-4-isoxazolepropionic acid glutamate receptor (AMPA glutamate receptor) transport at excitatory synapses. Thus, the molecular mechanisms in relation with the synaptic function of netrin-1 presents a new therapeutic target for neuropathology related to memory dysfunction including Alzheimer’s disease [206].

Several reports have additionally illustrated role of netrin–1′s role and its effect on AD pathology via Aβ production. Shabani et al. demonstrated that injection of netrin-1 in rats improved the amyloid-β-mediated suppression of learning-memory and synaptic plasticity. This study was included field potential recording and behavioral experimentation. They found that repeated administration of netrin–1 or its vehicle ameliorated neurotoxicity induced by Aβ and participated only in the recovery of the late phase of long-term potentiation (LTP) and did not contribute to the induction of early HFS-LTP (High frequency stimulation – long term potentiation) [207]. Likewise, Zamani et al. also investigated the effect of netrin-1 microinjection on memory impairment in a rat model of AD. They observed that in novel object recognition task (NOR), administration of netrin-1 enhanced cognitive dysfunction. At the same time, using the Morris water maze (MWM) setting, they also observed reduced spatial memory progression and reduced amyloid aggregation [208]. Nonetheless, netrin-1′s molecular mechanism associated with in spatial memory improvement and cognition is unclear. However, these studies provide a plausible therapeutic target for the treatment of AD in relation to the role of netrin–1 in AD pathology. These studies strongly support the implication of netrin signaling in AD pathogenesis. Netrin-1 consequently, serves as a complementary target for blocking neuronal cell death, ameliorating synaptic plasticity, and learning-memory functions induced by amyloid beta peptides.

The pathophysiology of AD is complex and one needs to consider several factors in comprehending the main component/s leading to memory loss, learning difficulties, poor judgement, and cognitive dysfunction. This review highlights the importance of receptors and neurotransmitters in synaptic signaling and cognitive function. This approach bridges the gap in our understanding of this enigmatic neurodegenerative disease, but also identifies prospective therapeutic targets to treat AD pathology. In this context, the pharamacological approaches and drug development methodologies for treating AD should be able to reverse the cognitive and synaptic impairments and reduce amyloid burden. While Aducanumab was recently approved by the FDA to remove Aβ plaques from the brains of AD patients, there is no definite proof that those AD patients will be able to regain their learning and memory functions. Consequently, defining the roles of receptors in synaptic transmission based on their significance in modulating nerve signals for a normal functioning brain, will enable researchers broaden understanding of AD pathology. Table 1 provides further insight of additional targets that have an essential role in cognitive functions.

Astrocytes are the non-neuronal cells that are involved in regulating neuronal health and blood brain barrier (BBB). They contribute towards neuronal development and homeostatic maintenance of neurotransmitters, glutamate and GABA [209]. They also serve as a target for ganglioside GM1 for mediating its cerebral energy metabolism and neuroprotective effect [210]. Thus, a dysfunctional astrocyte is characterized with a wide variety of neurodegenerative disease, such as AD, Parkinson’s disease, Huntington’s disease and Rhett syndrome [209]. In AD, reactive astrocytes are closely linked with Aβ peptides. They are considered as a direct modulator of synaptic transmission and neuronal functioning, resulting into impaired cognition in AD. Thus, targeting reactive astrocytes for reducing reactive astrogliosis is considered as a potential therapeutic approach for ameliorating cognition in AD [211]. Similarly, the astrocytic circadian clock is considered as a regulator for inducing the inflammation of Chi3l1 protein. Chi3l1/YKL-40 is a glycoprotein that is encoded by Chi3l1 gene. It is mainly expressed in astrocytes in the brain and is elevated during the inflammation of neurons in AD. Chi3l1/YKL-40 is also a potential therapeutic target for slowing the progression of AD and modulating the activation of glial phagocytosis and amyloid deposition in mice [212].

Low density lipoprotein receptor (LDLR) regulates the amount of cholesterol in the blood. It consists of cell surface proteins that are involved in endocytosis of plasma lipoprotein containing ApoE. The overexpression of LDLR have been reported in ameliorating cognitive decline by reducing tauopathy [213]. The expression of LDLR is modulated by Idol, an E3 ubiquitin ligase. The down regulation of Idol increases the concentration of LDLR in brain with the subsequent reduction in the amount of ApoE, amyloid plaques and finally ameliorating neuroinflammation via LXR-Idol pathway [214]. Thus, LDLR in relation to ApoE is crucial in removing plaques and tangles and serves as a potential therapeutic target for treating AD pathogenesis. 

Notch receptors (notch signaling) are highly expressed in the hippocampal area. These are instrumental in vascular development and also in determining the fates of neuronal and non-neuronal cell in development. The expression level of notch receptors changes in dementia patients and are closely associated with the risk of development of AD pathogenesis [215,216,217]. Likewise, the collagen receptor glycoprotein VI (GPVI) was reported to be involved in the promotion of platelet mediated amyloid aggregation in cerebral vessel through integrin αIIbβ3. Thus, inhibiting such processes may improve the vascular symptoms and cerebral amyloid angiopathy that contributes to dementia in AD patients [218,219]. Similarly, the elimination of orphan G-protein coupled receptor (GPR3) that mediates the amyloidogenic proteolysis significantly diminished the amyloid plaques and ameliorated memory in the transgenic AD mouse models [220]. And, in recent studies, the epigenetic modification, histone methylation, have been implicated in aging and neurodegenerative disorders. The epigenetic mechanism is crucial for learning and memory processes and serves as a possible therapeutic approach for AD pathogenesis [221,222].

## 11. Conclusions

Alzheimer’s disease (AD) is the most common neurodegenerative disorder, affecting a rapidly growing aging population throughout the world. In general, AD is characterized by the presence of extracellular amyloid plaques and intracellular NFTs and for this reason most studies have targeted plaque and tangle removal from the AD brain to treat this pervasive disease. After a century of disease identification efforts and 50 years of genetic and neurological studies aimed at identifying a cure, only a handful of drugs have been recognized and used as a possible cure. Treatment processes involve either symptomatic approaches to reduce memory loss and behavioral impairment, or more recently, disease-modifying approaches to delay clinical decline The FDA has recently approved the use of aducanumab, a monoclonal antibody, that is claimed to be capable of directly removing the amyloid plaques and restoring both cognition and memory functions in people living with AD. Aducanumab is the first AD therapeutic that attempts to treat the possible cause of the disease, and not just the symptoms. Nonetheless, the mechanistic relationship between the amyloid plaques and neurodegeneration remains ill defined and lacks unanimous confirmation from the scientific community guaranteeing safe administration of this drug to AD patients. 

AD is a progressive neurodegenerative disease characterized by memory impairments, behavioral changes, and reduced thinking capabilities. Patients experience confusion about places, time, and events and eventually are unable to carry out the simplest tasks. The underlying cause of the brain disorder associated with AD pathology remains ambiguous, with the possible involvement of a many neurotransmitters, their receptors, and signaling mechanisms. Distinct receptors have a definite role in relaying the nerve signals for the brain’s proper functioning in terms of cognitive function, learning, memory performance, and behavioral abilities. It is imperative that the treatment of AD narrows to a multifactorial approach where each subject of interest, whether molecular genes, receptor proteins, or signaling mechanisms, should be pursued separately to identify therapeutic targets to treat AD pathology. 

## Figures and Tables

**Figure 1 ijms-22-12987-f001:**
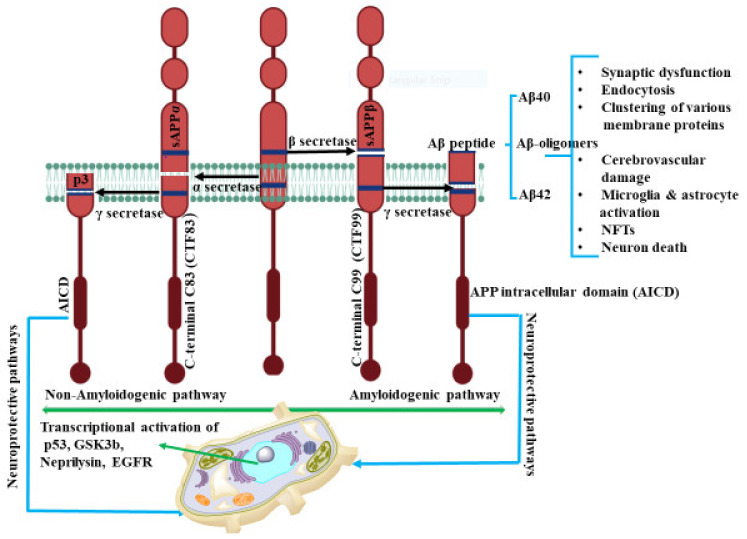
The amyloidogenic and non-amyloidogenic pathways. In non-amyloidogenic pathways, the APP cleavage occurs by α-secretase and produces CTF83 and sAPPα. sAPPα promotes neurite outgrowth, synaptogenesis, and cell adhesion. Further γ cleavage of CTF83 leads to the formation of AICD and p3. However, in the amyloidogenic pathway, β-secretase cleaves APP and results in formation of CTF99 and sAPPβ. γ cleavage of CTF99 leads to the formation of AICD and Aβ peptide. AICD generated by γ-secretase cleavage of CTF83 or CTF99 plays a role in nuclear translocation and transcriptional activation of target genes such as p53, GSK3β, neprilysin, EGFR. The biological relevance of CTF83 and CTF99 generated by α- and β-secretase, respectively, is unknown. Aβ peptide results in synaptic impairment and destroys the integrity of brain functions through various progressions. The figure was created using tools obtained from BioRender.com.

**Figure 2 ijms-22-12987-f002:**
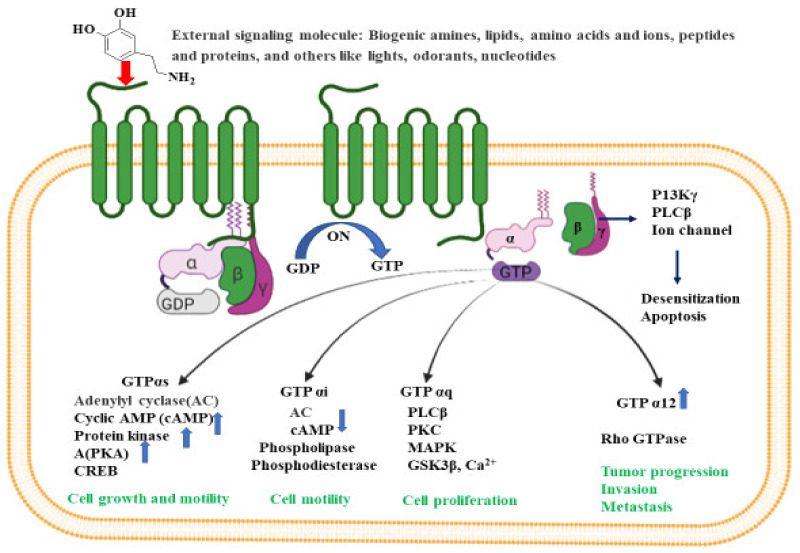
Intracellular signaling pathways for GPCR. Inactive GPCR exists as a complex unit consisting of α subunit bound to GDP and the βγ subunits. The binding of external signal molecules to a GPCR makes it active and results in a conformational change. During conformation, the GDP of α subunits forms to GTP, and the separation of α and βγ subunits occurs. When an active Gα subunit dissociates from the GPCR, it may interact with other proteins, triggering a signaling pathway that leads to cell growth and motility, cell proliferation, tumor progression, invasion, metastasis etc., depending on the binding proteins. The figure was created using tools obtained from BioRender.com.

**Figure 3 ijms-22-12987-f003:**
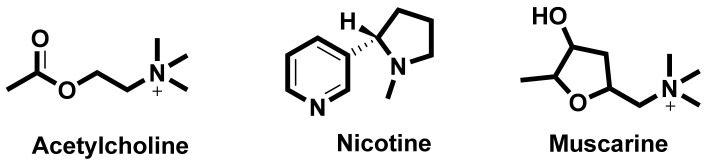
Key molecules comprising the cholinergic system.

**Figure 4 ijms-22-12987-f004:**
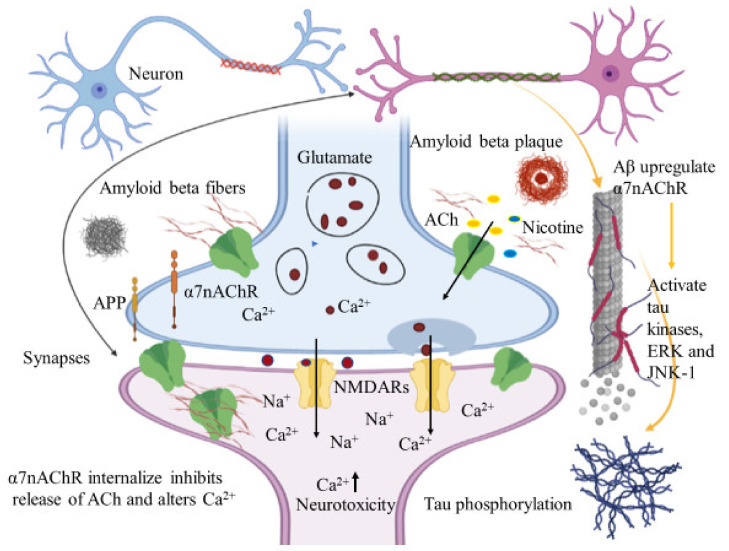
Amyloid beta interacts in a futile cycle with α7nAChRs. Amyloid beta interacts with α7nAChRs increasing the flux of intracellular Ca^2+^ in the presynaptic region. Ca^2+^ releases glutamate, resulting in postsynaptic depolarization followed by NMDAR activation. Long-term activation desensitizes and internalizes NMDARs and α7nAChRs post synaptically, inhibits the release of ACh, and alters Ca^2+^. In addition, Aβ up regulates α7nAChRs and induces tau phosphorylation via the activation of tau kinases, ERK, and JNK-1. The figure was created using tools obtained from BioRender.com.

**Figure 5 ijms-22-12987-f005:**
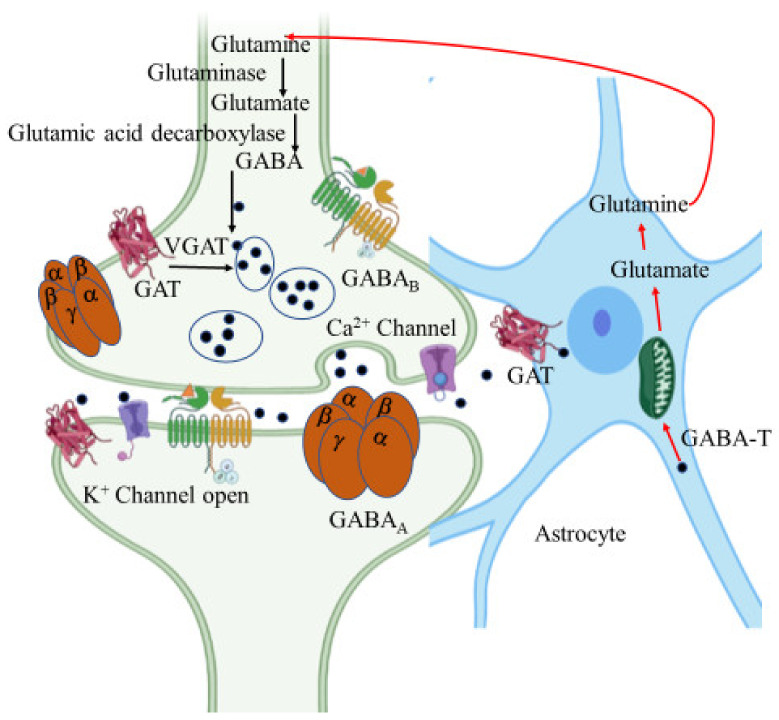
γ-Aminobutyric acid (GABA) synthesis pathway. GABA is synthesized from glutamate by glutamic acid decarboxylase in GABAergic neurons. It is then packed into the synaptic vesicles by GABA transporter (GAT) and then released into the synapse or synaptic cleft by membrane depolarization. Released GABA binds with GABA_A_ and GABA_B_ receptors resulting in inhibition of the postsynaptic neuron. In astrocytes, GABA is recycled into synaptic vesicles or taken up by mitochondria, where it is metabolized by GABA transaminase (GABA-T) to glutamine for neuronal uptake. The figure was created using tools obtained from BioRender.com.

**Figure 6 ijms-22-12987-f006:**
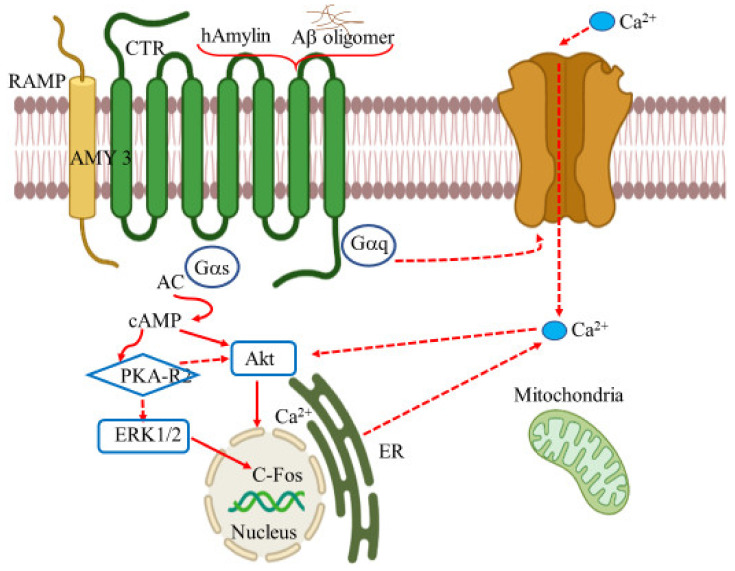
The AMY_3_ is a heterodimeric complex of calcitonin receptor and receptor activity modifying protein 3. In AD, Amyloid β (Aβ) peptide activates AMY_3_ subtype receptors by promoting multiple intracellular signaling pathways which is demonstrated in the above figure. The interaction of Human amylin (hAmylin) and Aβ together leads to activation AMY_3._ The activation of AMY3 stimulates G-Protein Gαs which further control the adenylate cyclase (AC), followed by an increase in cellular cAMP. Binding of cAMP to the PKA regulatory subunits (R_2_) induces dissociation of the tetrameric PKA holoenzyme, resulting in activation of PKA catalytic subunits. cAMP-activated PKA is involved in the regulation of Erk1/2 activities. The triggering of the ERK1/2 pathway may cause uneven distribution of Ca^2+^ trigger the dysfunction of the endoplasmic reticulum (ER) and mitochondria and result in cell death. In addition, cAMP stimulates Akt which further leads to transcription factor cFos expression. Created with BioRender.com.

**Table 1 ijms-22-12987-t001:** Promising therapeutic targets for ameliorating AD.

S.N.	Potential Target	Therapeutic Approach & Observation	Conclusion	Ref. No.
**1.**	**Astrocytes**			
	**Reactive astrocytes** Astrocytes are non-neuronal cells in CNS that are involved in regulating the neuronal health and blood-brain barrier (BBB) function. These astrocytic cells serve as a target for ganglioside GM1 for mediating its cerebral energy metabolism and neuroprotective effects.	Reactive astrocytes are closely linked with Aβ peptides and may regulate synaptic transmission and function of neuronal network, thereby resulting in impaired cognitive function in AD. Modifying the reactive astrocytes in the APPswePS1dE9 AD mouse model influences cognition and AD pathogenesis.	In AD, targeting reactive astrocytes characterized by enhanced intermediate filament proteins and cellular hypertrophy, to reduce astrogliosis is effective in ameliorating cognition.	[210,211,212]
	**Chi3l1/YKL-40** YKL-40, a glycoprotein encoded by the Chi3l1 gene, is a human CSF biomarker of neuro inflammation, which is elevated in AD.	Deletion of Chi3l1 decreased amyloid plaque burden and increased periplaque expression of the microglial lysosomal marker CD68 in the APP/PS1 mouse model of AD.	Chi3l1/YKL-40 regulates glial activation, Aβ phagocytosis, and amyloid plaque deposition in mice and influences AD progression in humans, suggesting that the astrocyte circadian clock regulates neuro inflammation as induced by Chi3l1.	[212]
**2.**	**Low-density lipoprotein receptor (LDLR) in relation to Apolipoprotein E (ApoE)**			
	**LDLR** LDLR is an ApoE metabolic receptor with a key role in cholesterol metabolism.	In P301S tauopathy mice, over expression of LDLR in microglia cells down regulated ApoE levels, resulting in suppressed microglial activation. Likewise, the reduced level of ApoE and increased level of LDLR favors microglial catabolism over anabolism and enhances the oligodendrocyte progenitor cells (OPCs) along with preserving myelin integrity.	Raising levels of LDL protein significantly reduced ApoE level in mouse brain and improved tau pathology and neurodegeneration.	[213]
	**Idol, an E3 ubiquitin ligase** Idol is an E3 ubiquitin ligase that is transcriptionally regulated by LXRs (liver X receptors), targeting LDLR for degradation.	Idol is responsible for metabolism of brain ApoE and Aβ plaque biogenesis. The down regulation of Idol expression in APP/PS1 mouse model of AD increases brain LDLR, decreases ApoE, and reduces soluble and insoluble Aβ peptides and amyloid plaque burden thereby improving neuro inflammation.	LXR-Idol pathways play a significant role in modulating LDLR and ApoE protein expression in brain and may affect AD pathogenesis involving the removal of apolipoprotein E and amyloid beta in the brain.	[214]
**3.**	**Notch Receptors** Notch receptors are transmembrane proteins consisting of epidermal growth factor in extracellular domain with a key role in vascular development and angiogenesis. These proteins are highly expressed in the hippocampal area (region of synaptic plasticity) and depends upon γ-secretase for its proteolytic functioning.	Patients suffering from dementia had low plasma levels of soluble notch 1 receptor, compared to their healthy counterparts. Following amyloid beta treatment, the level of notch 1 protein and notch 1 mRNA level increased remarkably in human brain microvascular endothelial cells (HBMECs) and human iPSC-derived neuronal cells.	The levels of notch 1 receptor vary significantly in AD patients and are considered to be involved in AD pathogenesis and vascular dementia.	[215,216,217]
**4.**	**Integrin α_IIb_β3 and Collagen receptor glycoprotein VI (GPVI)**	Blocking the binding pathways and stimulating Aβ with integrin α_IIb_β3 and GPVI may therapeutically reduce amyloid plaque formation mediated by platelet in cerebral vessels and brain parenchyma of AD patients.	Inhibition of integrin α_IIb_β3 and GPVI on the surface of platelets may ameliorate vascular symptoms and cerebral amyloid angiopathy that contributes to dementia in AD patients.	[218,219]
**5.**	**GPR3** GPR3, the orphan G-protein coupled receptor, modulates the function of γ-secretase and the generation of Aβ peptide in the absence of Notch receptor proteolysis.	In four AD transgenic mouse models (a single APP transgenic model, a double APP/PS1transgenic model, and two App knock-in transgenic mouse models), the genetic deletion or loss of GPR3 decreased amyloid pathology in all of the models and alleviated cognitive deficits in the APP/PS1 mice.	GPR3 mediates the amyloidogenic proteolysis of APP. GPR3 removal significantly diminished the amyloid plaques and ameliorated memory in the transgenic AD mouse models.	[220]
**6.**	**Histone H3 lysine K4 trimethylation (H3K4me3)** The transcriptional regulation of H3K4 methylation has been implicated in hippocampal and striatum-dependent memory formation in mice and human cognitive impairment.	Inhibition of H3K4-specific methyltransferases (catalyzed Histone H3K4me3 enzyme) in the P301S tau transgenic mouse model, significantly improved glutamatergic synaptic function and memory in PFC (pre-frontal cortex) pyramidal neurons.	Treatment of P301S mutant tau mouse model with a specific Sgk1 inhibitor, significantly reduced hyper phosphorylated tau protein in the frontal cortex and recovered the glutamatergic synaptic transmission in the mouse model, indicating the importance of H3K4me3-mediated Sgk1 up-regulation association of with AD-related pathologies.	[221,222]
	**Sgk1 (serum and glucocorticoid-regulated kinase 1) gene** Sgk1 gene encodes serum and glucocorticoid-regulated kinase 1, and is highly expressed in PFC of AD patients.	Inhibition of the up-regulated levels of Sgk1 in P301S Tau model mice by the use of a specific Sgk1 inhibitor leads to the reduction of hyperphosphorylated tau protein, along with restoration of PFC glutamatergic synaptic function, and improvement of memory impairments in AD mice.		

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
