# Peer review of "Role of Receptors in Relation to Plaques and Tangles in Alzheimer’s Disease Pathology"

_ijms, 2021, doi:10.3390/ijms222312987_

Round 1
Reviewer 1 Report
This review covers a wide range of receptors that may play roles in AD pathology and be potential therapeutic targets.
I have the following thoughts about this review:
- About half of the items from Table1 are not receptors, which is a deviation off the title and the topic of this article. The rest are receptors, so why not elaborate them in the main text?
- AMPA receptors are the major excitatory neurotransmission mediators, and their dysregulation and related synaptic dysfunction is an important part of AD etiology and pathology, which may earn them a place in this review.
- The new nomenclature is recommended for NMDA receptor subunits such as GluN2B.
Author Response
Reviewer #1
Comment 1
About half of the items from Table1 are not receptors, which is a deviation off the title and the topic of this article. The rest are receptors, so why not elaborate them in the main text?
Response 1
Thank you for your suggestion. We would like to clarify that in “Table 1”, we have included the possible targets for ameliorating AD that may or may not be the receptors. Our aim was to enlist the additional therapeutic targets of AD in brief that have essential role in cognitive function, in order to make the readers aware of the other existing and promising targets. Nonetheless, we have elaborated in the main text as:
Astrocytes are the non-neuronal cells that are involved in regulating neuronal health and blood brain barrier (BBB). They contribute towards neuronal development and homeostatic maintenance of neurotransmitters, glutamate and GABA [210]. They also serve as a target for ganglioside GM1 for mediating its cerebral energy metabolism and neuroprotective effect [211]. Thus a dysfunctional astrocyte is characterized with a wide variety of neurodegenerative disease, such as AD, Parkinson’s disease, Huntington’s disease and Rhett syndrome [210]. In AD, reactive astrocytes are closely linked with Aβ peptides. They are considered as a direct modulator of synaptic transmission and neuronal functioning, resulting into impaired cognition in AD. Thus, targeting reactive astrocytes for reducing reactive astrogliosis is considered as a potential therapeutic approach for ameliorating cognition in AD [212]. Similarly, the astrocytic circadian clock is considered as a regulator for inducing the inflammation of Chi3l1 protein. Chi3l1/YKL-40 is a glycoprotein that is encoded by Chi3l1 gene. It is mainly expressed in astrocytes in the brain and is elevated during the inflammation of neurons in AD. Chi3l1/YKL-40 is also a potential therapeutic target for slowing the progression of AD and modulating the activation of glial phagocytosis and amyloid deposition in mice [213].
Low density lipoprotein receptor (LDLR) regulates the amount of cholesterol in the blood. It consists of cell surface proteins that are involved in endocytosis of plasma lipoprotein containing ApoE. The overexpression of LDLR have been reported in ameliorating cognitive decline by reducing tauopathy [214]. The expression of LDLR is modulated by Idol, an E3 ubiquitin ligase. The down regulation of Idol increases the concentration of LDLR in brain with the subsequent reduction in the amount of ApoE, amyloid plaques and finally ameliorating neuroinflammation via LXR-Idol pathway [215]. Thus, LDLR in relation to ApoE is crucial in removing plaques and tangles, and serves as a potential therapeutic target for treating AD pathogenesis.
Notch receptors (notch signaling) are highly expressed in the hippocampal area. These are instrumental in vascular development and also in determining the fates of neuronal and non-neuronal cell in development. The expression level of notch receptors changes in dementia patients and are closely associated with the risk of development of AD pathogenesis [216-218]. Likewise, the collagen receptor glycoprotein VI (GPVI) was reported to be involved in the promotion of platelet mediated amyloid aggregation in cerebral vessel through integrin αIIbβ3. Thus inhibiting such processes may improve the vascular symptoms and cerebral amyloid angiopathy that contributes to dementia in AD patients [219,220]. Similarly, the elimination of orphan G-protein coupled receptor (GPR3) that mediates the amyloidogenic proteolysis significantly diminished the amyloid plaques and ameliorated memory in the transgenic AD mouse models [221]. And, in recent studies, the epigenetic modification, histone methylation, have been implicated in aging and neurodegenerative disorders. The epigenetic mechanism is crucial for learning and memory processes and serves as a possible therapeutic approach for AD pathogenesis [222,223].
Comment 2
AMPA receptors are the major excitatory neurotransmission mediators, and their dysregulation and related synaptic dysfunction is an important part of AD etiology and pathology, which may earn them a place in this review.
Response 2
Thank you for your suggestion. We have included AMPA receptors in the review with the title “AMPA receptor”;
α-amino-3-hydroxy-5-methyl-4-isoxazolepropionic acid receptor (AMPAR)
AMPA receptors are the ionotropic glutamate receptors formed by the combination of four subunits, GluA1-4 that are encoded by GRIA1-4 genes. These glutamate receptors mediate a rapid post synaptic transmission in CNS with a major role in long term synaptic plasticity such as LTP and LTD, and are also involved in long term memory retrieval (1). The biophysical property of AMPARs, involving the subunit composition, is instrumental in understanding the mode of AMPAR trafficking and synaptic plasticity. The most recurring AMPAR is the heteromeric composition of GluA1A2, which is dominant at CA1 cell synapses and is the primary mediator of synaptic plasticity (2). And, similarly, the dysfunction in the expression of all the four subunits of AMPAR, have also been reported as a cause of neurodevelopmental disorders (NDDs) (3). Thus, in this instance, it is important to highlight the involvement of AMPARs in the development of AD pathology, given its critical role in synaptic transmission and NDDs.
AMPARs are considered as a susceptible target of AD pathophysiology. The high levels of Aβ impairs the functional AMPAR and disrupts the excitatory synaptic transmission (4). Where, it has been reported that the cognitive impairment involves an age related downscaling of post synaptic AMPAR function in double knock in mice (2 X KI) with human mutation of gene for APP and presenilin (5). The reduction of AMPAR indicates that there are changes in the early phase of pathological molecular level in AD. In this regard, Zhang et. al. suggested that the reduction in the level of AMPAR is dependent on amyloid beta as Aβ are responsible for the ubiquitination and degradation of AMPAR in primary neurons, and finally, suppressing the synaptic transmission in AD (6). Further, the soluble form of Aβ oligomers have also been found to be involved in synaptic degradation and cognitive deficits that takes place via the subunit GluA3 AMPAR, indicating the crucial role of GluA3 in Aβ mediated degradation of synapse and cognitive functions (7). Thus, in this scenario, it is highly crucial to implement the drugs that target the elevation of AMPARs for alleviating memory decline and cognitive impairment in AD.
References:
- (a) Gasbarri, A.; Pompili, A. Involvement of Glutamate in Learning and Memory. Identification of Neural Markers Accompanying Memory, 2014, 63-77. (b) Pereyra, M.; Medina, J. H. AMPA Receptors: A Key Piece in the Puzzle of Memory Retrieval. Frontiers in human neuroscience, 2021, 15, 729051. (c) Henley, J.; Wilkinson, K. Synaptic AMPA receptor composition in development, plasticity and disease. Nat Rev Neurosci. 2016, 17, 337–350.
- Lu, W.; Shi, Y.; Jackson, AC; Bjorgan, K; During, MJ; Sprengel, R.; Seeburg, PH; Nicoll, RA. Subunit composition of synaptic AMPA receptors revealed by a single-cell genetic approach. , 2009, 62(2), 254-268.
- (a) Geisheker, M. R. et al. Hotspots of Missense Mutation Identify Neurodevelopmental Disorder Genes and Functional Domains. Nat. Neurosci., 2017, 20, 1043–1051. (b) Salpietro, V.; Dixon, C.L.; Guo, H. et al. AMPA receptor GluA2 subunit defects are a cause of neurodevelopmental disorders. Nat Commun., 2019, 10, 3094. (c) Davies, B. et al. A point mutation in the ion conduction pore of AMPA receptor GRIA3 Causes Dramatically Perturbed Sleep Patterns as well as Intellectual Disability. Hum. Mol. Genet., 2017, 26, 3869–3882. (d) Martin, S. et al. De-novo variants in GRIA4 Lead to Intellectual Disability with or without Seizures and Gait Abnormalities. Am. J. Hum. Genet., 2017, 101, 1013–1020.
- Guntupalli, S.; Widagdo, J.; Anggono, V. Amyloid-β-Induced Dysregulation of AMPA Receptor Trafficking. Neural Plast., 2016, 3204519.
- Chang, E.H.; Savage, M.J.; Flood, D.G.; Thomas, J.M.; Levy, R.B.; Mahadomrongkul, V.; Shirao, T. Aoki, C.; Huerta, P.T. AMPA receptor downscaling at the onset of Alzheimer's disease pathology in double knockin mice. Proc Natl Acad Sci, 2006, 103(9), 3410-3415.
- Zhang, Y.; Guo, O.; Huo, Y.; Wang, G.; Man, H.Y. Amyloid-β Induces AMPA Receptor Ubiquitination and Degradation in Primary Neurons and Human Brains of Alzheimer's Disease. J Alzheimers Dis. 2018, 62(4), 1789-1801.
- Reinders, N. R; Pao, Y.; Renner, M. C.; da Silva-Matos, C. M.; Lodder, T. R.; Malinow, R.; Kessels, H. W. Effects of Amyloid-β Require AMPAR Subunit GluA3. Proc Natl Acad Sci, 2016, 113 (42), E6526-E6534.
Comment 3
The new nomenclature is recommended for NMDA receptor subunits such as GluN2B.
Response 3
Thank you for your suggestion. We have changed accordingly, as shown below:
N-methyl-D-aspartate receptor (NMDAR) is a ligand gated ionotropic glutamate receptor that selectively binds with NMDA for neurotransmission. These glutamatergic receptor consists of two glycine-binding subunits (i.e. GluN1), two glutamate-binding subunits (i.e., GluN2A, GluN2B, GluN2C, and GluN2D), a combination of a GluN2 subunit and glycine-binding GluN3 subunit (i.e., GluN3A or GluN3B), or two GluN3 subunits. Thus, in NMDR, the binding of two different neurotransmitters, glutamate and glycine is essential for the activation of glutamate-gated ion channel [61].
Reference
- Yamamoto, H.; Hagino, Y.; Kasai, S.; Ikeda, K. Specific Roles of NMDA Receptor Subunits in Mental Disorders. Current molecular medicine, 2015, 15(3), 193–205.
Reviewer 2 Report
Dear authors,
This review paper nicely desbribes the relationship of verious receptors and Alzhimer's disease pathology. Since recent findings suggest that alteration in neuronal activity is involved in Alzheimer's disease pathogenesis, this kind of review paper will surely help readers to get current information.
The sections of NMDAR, cholinergic receptors, and GABAR are quite important. The authors should include those receptors.
Author Response
Reviewer #2
Comment 1
The sections of NMDAR, cholinergic receptors, and GABAR are quite important. The authors should include those receptors.
Response 1
Thank you for your suggestion. However, we would like to notify that NMDAR, cholinergic receptors and GABA receptors have already been included in the manuscript.